# Revealing the immune perturbation of black phosphorus nanomaterials to macrophages by understanding the protein corona

Jianbin Mo [1,2], Qingyun Xie[3], Wei Wei [1,2] & Jing Zhao [1]

The increasing number of biological applications for black phosphorus (BP) nanomaterials has precipitated considerable concern about their interactions with physiological systems. Here we demonstrate the adsorption of plasma protein onto BP nanomaterials and the subsequent immune perturbation effect on macrophages. Using liquid chromatography tandem mass spectrometry, 75.8% of the proteins bound to BP quantum dots were immune relevant proteins, while that percentage for BP nanosheet–corona complexes is 69.9%. In particular, the protein corona dramatically reshapes BP nanomaterial–corona complexes, influenced cellular uptake, activated the NF-κB pathway and even increased cytokine secretion by 2–4-fold. BP nanomaterials induce immunotoxicity and immune perturbation in macrophages in the presence of a plasma corona. These findings offer important insights into the development of safe and effective BP nanomaterial-based therapies.

[1] State Key Laboratory of Coordination Chemistry, Institute of Chemistry and BioMedical Sciences, School of Chemistry and Chemical Engineering Nanjing University, Nanjing 210093, China. [2] State Key Laboratory of Pharmaceutical Biotechnology, School of Life Sciences, Nanjing University, Nanjing 210093, China. [3] Department of Orthopedics, Chengdu Military General Hospital, Chengdu 610083, China. These authors contributed equally: Jianbin Mo, Qingyun Xie. Correspondence and requests for materials should be addressed to W.W. (email: weiwei@nju.edu.cn) or to J.Z. (email: jingzhao@nju.edu.cn)

Black phosphorus (BP) crystals have existed for a long time, but we have only recently begun to realize their true potential from the perspective of two-dimensional (2D) nanomaterials[1]. BP thin film material has been produced from bulk BP crystals through the exfoliation method, and this material exhibits unique qualities such as thickness-dependent band gap and high carrier mobility[2,3]. While many breakthroughs have been made regarding nanoelectronic applications for BP nano-materials[4], studies on extensive applications in biomedicine and biotechnology have become more common. Several groups have reported that BP nanomaterials could serve as high-efficiency photosensitizers for photodynamic therapy and drug delivery systems with ultra-high loading capacity[5–8]. Recently, potential inflammatory effects of BP nanomaterials have been reported[9]. However, understanding of their actual biological effects on physiological systems is limited.

It has now been established that the surfaces of nanomaterials are immediately covered by a plasma protein corona when they come into contact with blood[10], a finding that has led to the redefinition of the biological properties of nanoparticles (NPs)[11,12]. Chen et al. elegantly demonstrated binding of the bovine serum albumin (BSA) protein corona to Au nanorods and revealed reduced damage to cell membranes[13]. Dawson et al. have made pivotal progress on understanding the role of the protein corona in the loss of targeting capabilities of NPs[14].

The implications of nanomaterial–corona complexes are far reaching, suggesting that the biological impacts of nanomaterials on a physiological system cannot simply be linked to the nature of the nanomaterial alone. Therefore, for continued development of BP nanomaterials in biomedicine, studies on interactions between nanomaterials and plasma proteins and the influence on immune cells are urgent. Here we study the protein corona formed by exposing different sizes of BP nanomaterials to plasma proteins and demonstrated their immune perturbation effect on macro-phage (Fig. 1). Our results also highlight the importance of nanomaterial size on protein corona formation and could further redefine BP nanomaterials and the biological response to BP nanomaterials.

## Results

**Characterization of nanomaterials**. First, BP nanosheets (BPNSs) and BP quantum dots (BPQDs) were obtained according to reported methods[5,7] and analysed by transmission electron microscopy (TEM). From the TEM image results, BPNSs were found to be free-standing with a lateral size of approximately 300 nm, and BPQDs were found to be approximately 5 nm in size (Fig. 2a, b). According to a statistical TEM analysis of 100 BPQDs (Fig. 2b), the average lateral size of a BPQD is 5.7 ± 1.0 nm (values are expressed as the means ± SDs of triplicates). The BPQDs and BPNSs were also characterized by X-ray photoelectron spectro-scopy (XPS) and Raman spectroscopy. As shown in Fig. 2c, the similar prominent peaks from BPQDs and BPNSs can be attributed to one out-of-plane phonon mode ($A_g^1$) at 356.8 cm$^{-1}$ as well as two in-plane modes, $B_{2g}$ and $A_g^2$, at 432.3 and 459.6 cm$^{-1}$, respectively. Furthermore, similar survey XPS spectra for BPQDs and BPNSs document the similar surface chemistry of these two BP nanomaterials (Fig. 2d). Taken together, the data indicate that size is the key difference between BPQDs and BPNSs.

Numerous studies have shown visual images of the corona on nanospheres or nanorods, but nanosheet coronas have rarely been reported. Therefore, to provide additional qualitative support for the stability of the isolated BP–corona complexes, we present TEM images of BPQDs and BPNSs with protein corona (Fig. 2e, f). In BPNS–protein corona complexes, the surfaces of the BPNSs were strikingly different from their native forms, and the zeta-potential values of BPNSs decreased from −18.1 to −8.4 mV (Fig. 2i), suggesting adsorption of plasma proteins onto the BPNSs. The size of BPNS–corona nanomater-ials was also investigated by dynamic light scattering (DLS) analysis (Fig. 2g and Supplementary Table 1). The results showed that the average size of BPNSs changed from 338.4 ± 2.3 to 365.3 ± 5.9 nm. Unexpectedly, after the protein corona was formed on the surface of BPQDs, BPQDs were redefined from ultra-small nanosheets to spherical BPQD–corona complex NPs, and the NPs appeared to be bulky particles with a diameter of approximately 371.9 ± 9.1 nm (Fig. 2b, f). The zeta-potential values of BPQDs decreased from −20.6 to −6.4 mV, and the "wet" diameter increased from 5.6 ± 1.4 to 362.5 ± 5.6 nm (Fig. 2h, i and Supplementary Table 1). Energy dispersive X-ray spectroscopic (EDX) analysis further verified that BPQD was in the corona complex (Supplementary Figure 1).

**Identification of the plasma corona**. According to the literature, the formation of plasma proteins on the surfaces of nanomaterials can occur almost instantly after nanomaterials enter the blood[15]. The zeta-potential values for BP nanomaterials showed that the high initial values (−20.6 mV for BPQDs and −18.1 mV for

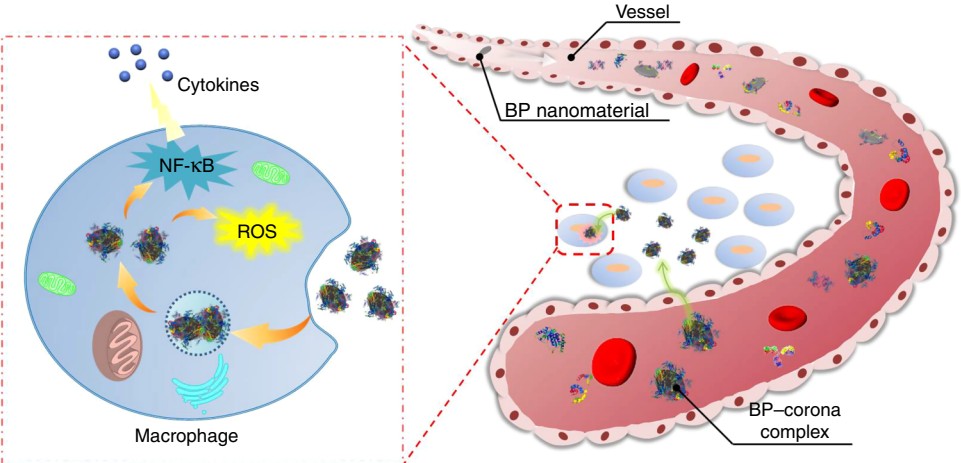

**Fig. 1** Summary diagram of immune perturbation of BP–corona complexes. Formation of BP nanomaterial–corona complexes in blood and their immunoregulation on macrophages

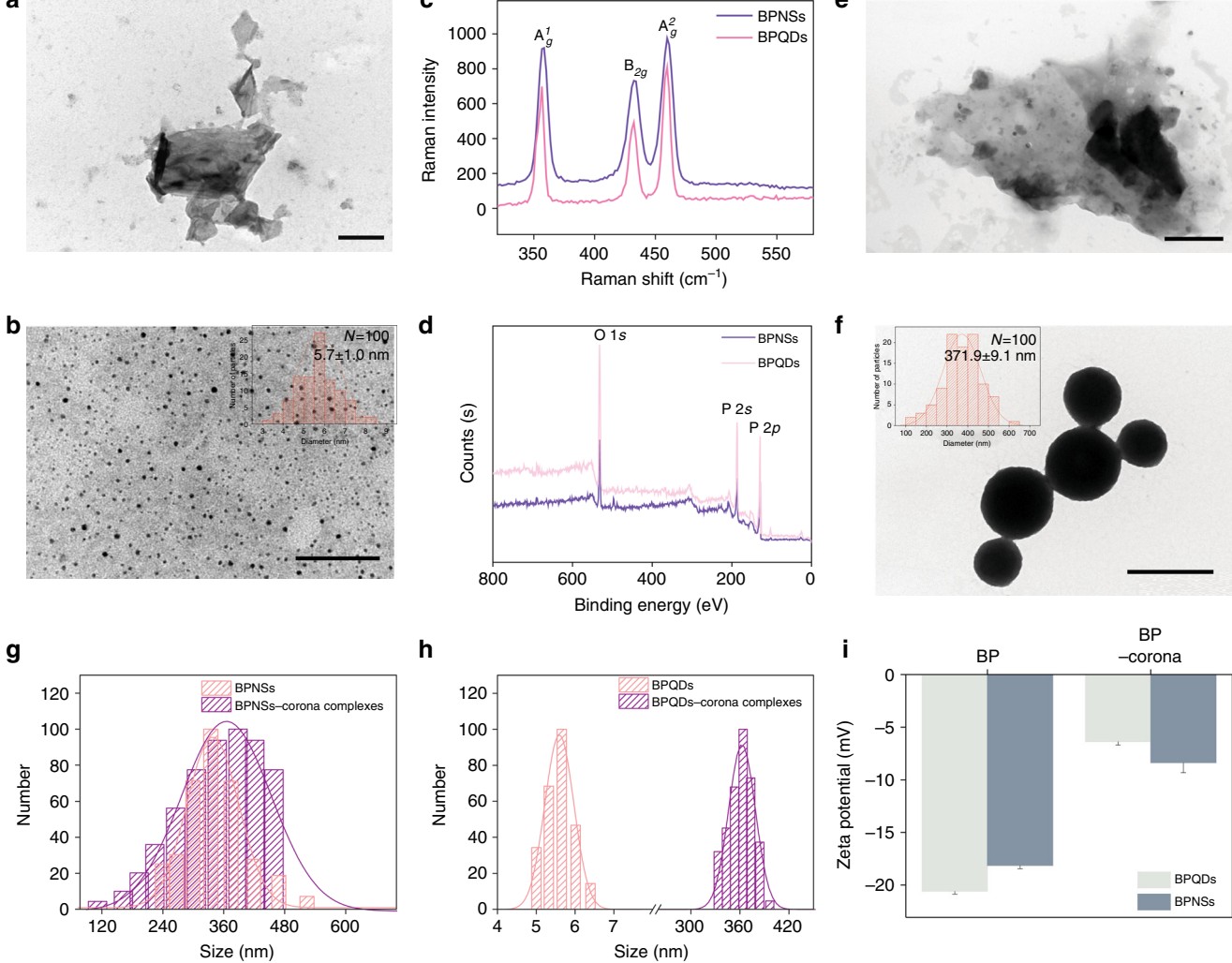

**Fig. 2** Image of BP nanomaterials and corona complexes. **a** TEM image of a BPNS. Scale bar: 200 nm. **b** TEM image of BPQDs. Inset: statistical analysis of the size of 100 BPQDs determined by TEM. Scale bar: 200 nm. **c** Raman spectra of BPQDs and BPNSs. **d** XPS spectra of BPQDs and BPNSs. **e** TEM image of a BPNS-corona complex. Scale bar: 50 nm. **f** TEM image of BPQD-corona complex. Inset: statistical analysis of the size of 100 BPQD-corona complexes determined by TEM. Scale bar: 500 nm. **g** Dynamic light scattering (DLS) analysis of BPNSs and BPNS–corona complexes. **h** DLS analysis of BPQDs and BPQD-corona complexes. **i** Zeta potential of BP nanomaterials and BP-corona complexes. Values are expressed as the means ± SDs of triplicates

BPNSs) were reduced after entry (to −7.3 mV for BPQDs and −13.7 mV for BPNSs in 0.5 h), suggesting that the proteins were adsorbed onto BP nanomaterials in a short time (Supplementary Figure 2). Then BP nanomaterials, including BPNSs and BPQDs, were exposed to blood plasma of different concentrations for 4 h. Plasma proteins in corona were then separated by sodium dodecyl sulphate-polyacrylamide gel electrophoresis (SDS-PAGE). As Fig. 3a shows, with an increasing plasma concentration, typical bands from BPNS–corona became more visible (i.e., more same type proteins were enriched at higher concentrations). Notably, we found that high-abundant proteins on BPQDs were changed when BPQDs interacted with plasma of different concentrations. It has been suggested that the formation of corona was affected significantly by nanomaterial size.[16,17]. The different protein-binding patterns from BPQD–corona in plasma of different concentrations revealed that competitive plasma proteins with an increasing adhesion at a higher concentration blood plasma could facilitate the desorption of proteins with lower binding affinity (Fig. 3b). Furthermore, we performed a more detailed study of the protein pattern to describe the observation associated with BPQD–corona. Thus, in Fig. 3c, d, we analysed the relative densitometry (the intensity of each band is divided by

the intensity of a fixed band, the 40 KD marker band in the first lane) of several bands marked with an asterisk from the gels in Fig. 3a, b. As shown in Fig. 3d, 55 and 60 KD proteins were main components in corona at low plasma concentration, while other proteins (i.e., 35, 70 and 180 KD protein) dominated the corona composition at high plasma concentration. These data suggested that the size of BP nanomaterials played a key role in determining the BP–corona complexes.

Next, we used liquid chromatography tandem mass spectrometry (LC-MS/MS) to qualitatively analyse components of the BP–protein corona (Supplementary Data 1, 2 and 3). Further analysis with LC-MS/MS revealed that, while BPNSs were able to adsorb 52 plasma proteins, BPQDs were capable of binding 96 proteins. Of these proteins, 16 proteins appeared on both BPNSs and BPQDs (Fig. 3e). Subsequently, we analysed the difference in protein component among plasma proteins, BPQD–corona complex proteins and BPNS–corona complex proteins. First, this analysis showed that the proteins on BPQDs and BPNSs do not simply correspond to the relative protein concentrations in blood plasma, as reported by other studies that focussed on other types of nanomaterials[16,18]. The bound proteins were further classified by gene ontology (GO) analysis according to their biological

process (Supplementary Figure 3). As shown in Fig. 3f, of the 186 proteins in blood, 73 proteins in BPQD–corona complexes and 36 proteins in BPNS–corona complexes were annotated as immune relevant proteins. These results indicated that 75.8% of the proteins bound to BPQDs (96 plasma proteins) were immune relevant proteins, while that percentage for BPNS–corona complexes (52 plasma proteins) is 69.9% (Fig. 3g). According to previous reports, negatively charged NPs primarily attract positively charged proteins. By contrast, we observed that overall, of the proteins bound to the negatively charged BP nanomaterials, more than half of the proteins (64.93% for BPNSs and 62.00% for BPQDs) had a negative charge (pI < 7.4), irrespective of their relative abundance in blood plasma (Supplementary Data 3 and Supplementary Figure 4). This result was also confirmed by zeta-potential measurements, revealing a

negative charge of BP–corona complexes (Supplementary Figure 2). Hence, an effective charge alone does not appear to be the major driving force regulating the BP–protein interaction. In addition, we discovered a significant enrichment of plasma proteins with a high molecular weight (MW) on BPNSs rather than on BPQDs, while proteins with a low MW were less enriched on BPNSs than BPQDs (Supplementary Figure 5). These results indicated a distinct, protein size-dependent particle-binding pattern.

**In vitro cytotoxicity assay and cellular uptake efficiency experiments**. Immune cells in the bloodstream (such as leukocytes, dendritic cells and monocytes) and tissues (such as macrophages) have a propensity to take up and clear certain nanomaterials[19]. Once plasma proteins are adsorbed onto BP

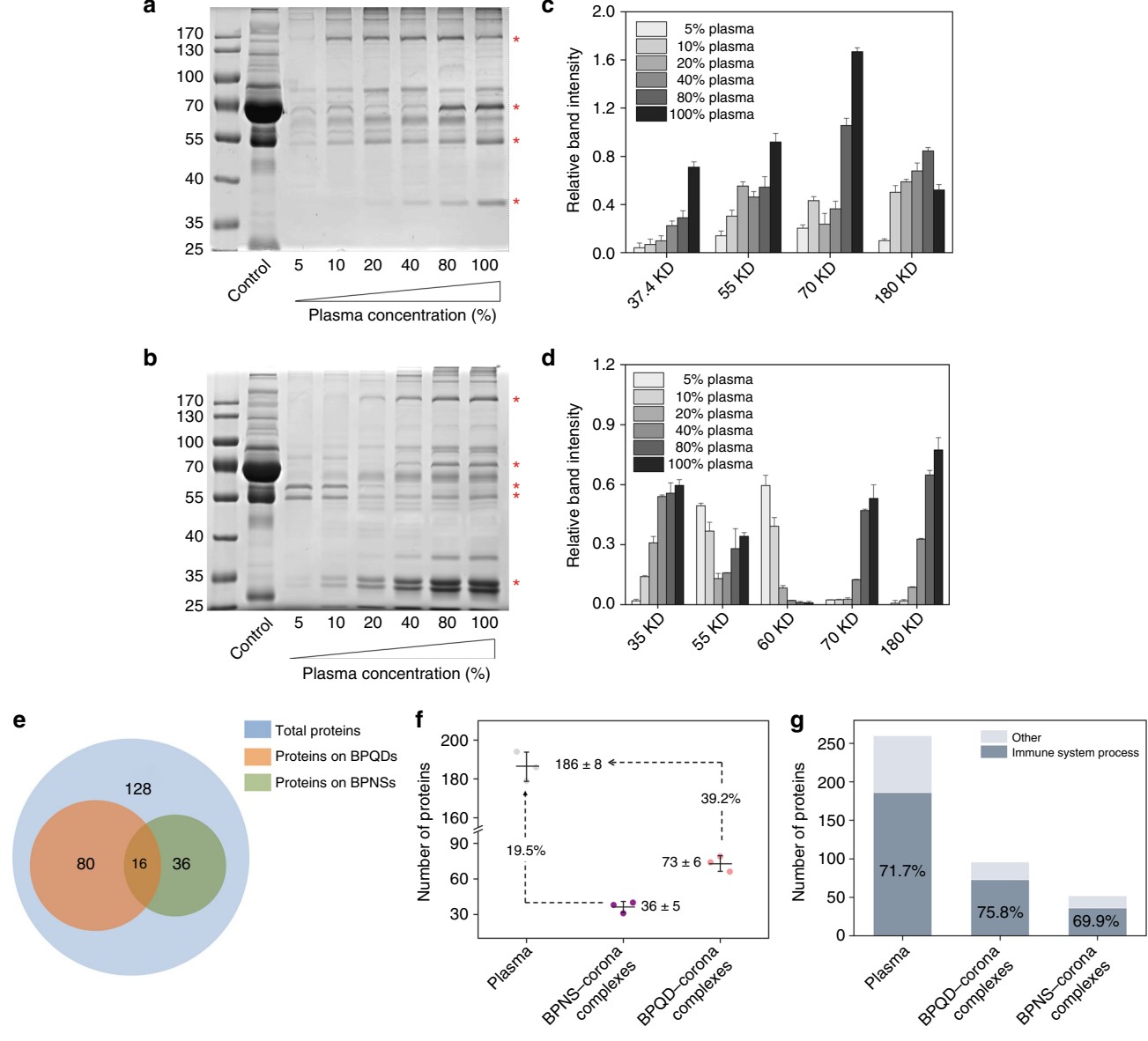

**Fig. 3** Corona protein identification and classification. SDS-PAGE analysis of plasma protein obtained from **a** BPNS–corona complexes and **b** BPQD–corona complexes. The molecular weights of proteins in the standard ladder are reported on the left. Relative amounts of the most abundant proteins bound to **c** BPNSs and **d** BPQDs as calculated by the grey-scale value from the gels shown in **a**, **b**. Values are expressed as the means ± SDs of triplicates. **e** Statistical analysis of the total number of proteins identified by LC-MS/MS and **f** the number of proteins on BPQDs and BPNSs involved in immune system processes according to gene ontology (GO) analysis. Values are expressed as the means ± SDs of triplicates. **g** Analysis of the immune relevant protein fraction

nanomaterials, some of these proteins could serve as opsonins to influence cellular uptake and clearance by immune cells and thus potentially affect distribution and delivery to the intended target sites. Since discovering the enrichment of immune proteins in BP–corona complexes, we further focussed on the immunomodulatory effects of BP nanomaterial–protein corona complexes in macrophages. In contrast to several studies that have reported nearly no cytotoxic effects of BP nanomaterials (including BPNSs and BPQDs) in cell lines, our data showed that BPNSs and BPQDs produced slight cytotoxic effects on cell lines[5,7], such as human lung cancer cells (H1299), human normal hepatic cells (L0-2), human normal kidney cells (293T), human macrophage-like cells (dTHP-1) and macrophages from peripheral blood (SC). Notably, when BPNSs and BPNSs were coated with the protein corona, the cytotoxic effect was significantly reduced (Supplementary Figure 6 and 7).

To further analyse the clearance of BP nanomaterials by macrophages, macrophage-like dTHP-1 cells were selected to evaluate the cellular uptake efficiency of BP–corona complexes.

As previous literature has shown, the formation of a corona on nanomaterials is a dynamic process, and the components of a human plasma corona are affected by foetal bovine serum (FBS)[20,21]. Thus it would be more suitable to carry out cell assays in serum-free media to eliminate the interference of FBS. To verify the effect of serum concentration on cells, we first performed an 3-[4,5-dimethylthiazol-2-yl]-2,5 diphenyl tetrazolium bromide (MTT) assay to analyse the cell viability in different culture media (0 and 10% FBS) after a short treatment. As Supplementary Figure 8a shows, after 6 h, the cell viabilities (without any treatment) are 98% in serum-free media and 100% in serum media. Even treated with nanomaterials (100 µg ml$^{-1}$) in serum-free media, the cell viabilities are all >90%. We next compared the cellular uptake of BP–corona complexes in different culture medium (0 and 10% FBS). The results of the cellular uptake experiment showed that FBS had no significant effect on cellular uptake efficiency during a short incubation period (Supplementary Figure 8b). Based on the above results, we carried out further measurements using macrophage-like cells in

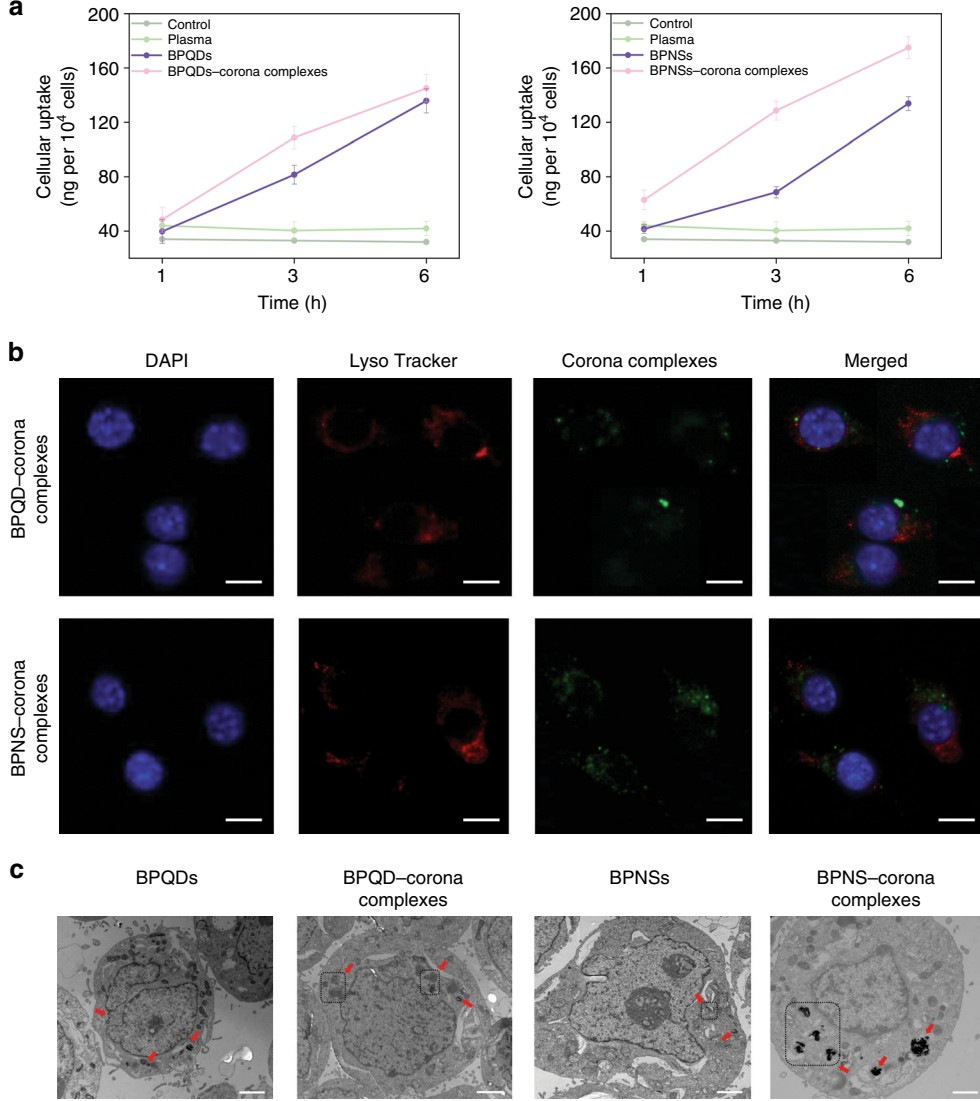

**Fig. 4** Cellular internalization of BP nanomaterials and corona complexes. **a** Cellular uptake of BP nanomaterials and BP–corona complexes. Macrophage-like dTHP-1 cells were treated with 150 µg ml$^{-1}$ nanomaterials for 1, 3 and 6 h. Cells treated with human plasma were set as the control. Values are expressed as the means ± SDs of triplicates. **b**, **c** Location of BP nanomaterials and BP–corona complexes in dTHP-1 cells. The cells were treated with 150 µg ml$^{-1}$ BP–corona complexes for 6 h and visualized using **b** fluorescence microscopy (scale bar: 10 µm) and **c** TEM (scale bar: 2 µm)

serum-free media to eliminate the interference of FBS. To clarify the association between the corona and cellular uptake, we next investigated the uptake efficiency of BPNSs, BPQDs and BP–corona complexes in dTHP-1 cells using inductively coupled plasma mass spectrometry (ICP-MS). As shown in Fig. 4a, the P content of cells incubated with BPQDs was higher (137.8 ng per $10^4$ cells after 6 h) than that of cells treated with BPNSs (130.8 ng per $10^4$ cells after 6 h), which was the reverse of the uptake pattern for graphene oxide nanosheets[22]. Typically, higher uptake levels of BP–corona complexes (175.0 ng per $10^4$ cells for BPNS–corona complexes and 145.1 ng per $10^4$ cells for BPQD–corona complexes) were detected after 6 h, indicating

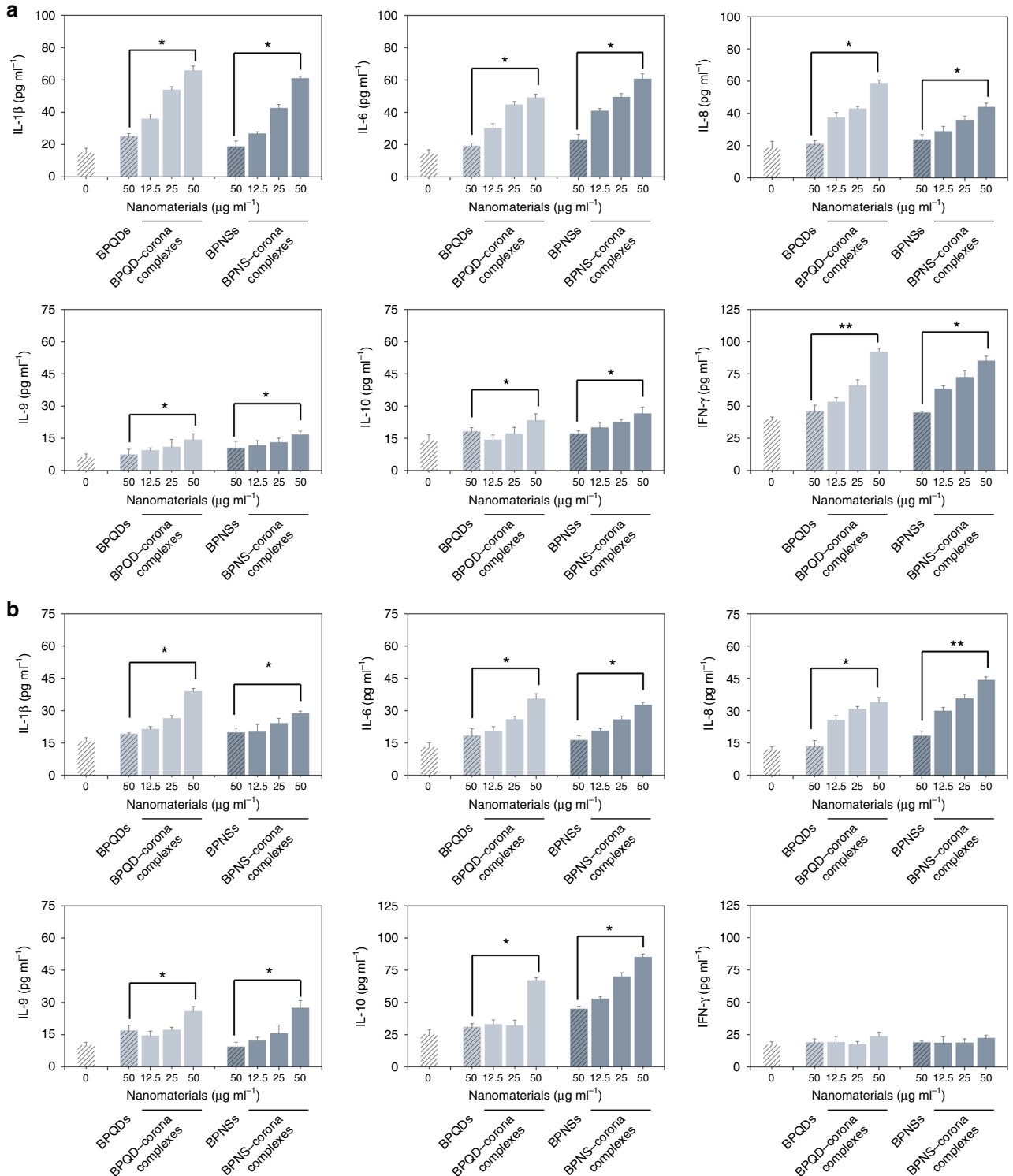

**Fig. 5** Cytokine secretion of different macrophages. Macrophage-like dTHP−1 cells (**a**) or SC cells (**b**) were treated with 50 µg ml$^{-1}$ BP nanomaterials and an increasing concentration of corona complexes (12.5, 25 and 50 µg ml$^{-1}$) for 6 h. Values are expressed as the means ± SDs of triplicates. Statistical significance is assessed by Student's $t$ test. *$p < 0.05$, **$p < 0.01$

the facilitation of BP nanomaterial internalization by the corona.

**Cellular locations.** To further investigate the intracellular behaviour, we added human serum albumin conjugated to fluorescein isothiocyanate (HSA-FITC) to plasma to create fluorescent corona complexes (Supplementary Figure 9). First, we studied the stability of fluorescent corona complexes in macrophage-like cell lysates to simulate the macrophage cell environment. As shown in Supplementary Figure 10, only 4.3% of HSA-FITC was dissociated from BPNS–corona complexes and 6.3% of HSA-FITC was dissociated from BPQD–corona complexes after 10 h, indicating the stability of fluorescent corona complexes in intracellular environments. In this way, we were able to label BP–corona complexes with fluorescence tags and determine their cellular location in dTHP-1 cells. Thus, fluorescence microscopy and TEM were combined to study the intracellular localization of BP–corona complexes. As shown in Fig. 4b, c, BP–corona complexes were internalized into macrophages by endocytosis, and the primary final location of the corona complexes was the cytoplasm. Moreover, the green fluorescence intensity of the BPNS–corona complexes was stronger than that of the BPQD–corona complexes, suggesting that BPNSs modified with plasma protein exhibited a superior ability to accumulate in macrophage-like cells compared with BPQDs. Nanomaterial size influences the efficiency of subsequent cellular uptake, probably by controlling the formation of the protein corona coating.

**Proinflammatory effect of BP–corona complexes in macrophages.** Presently, increasing studies have shown that protein corona formation was an important pathway for nanomaterials to stimulate the inflammatory response[23,24]. Other nanomaterial

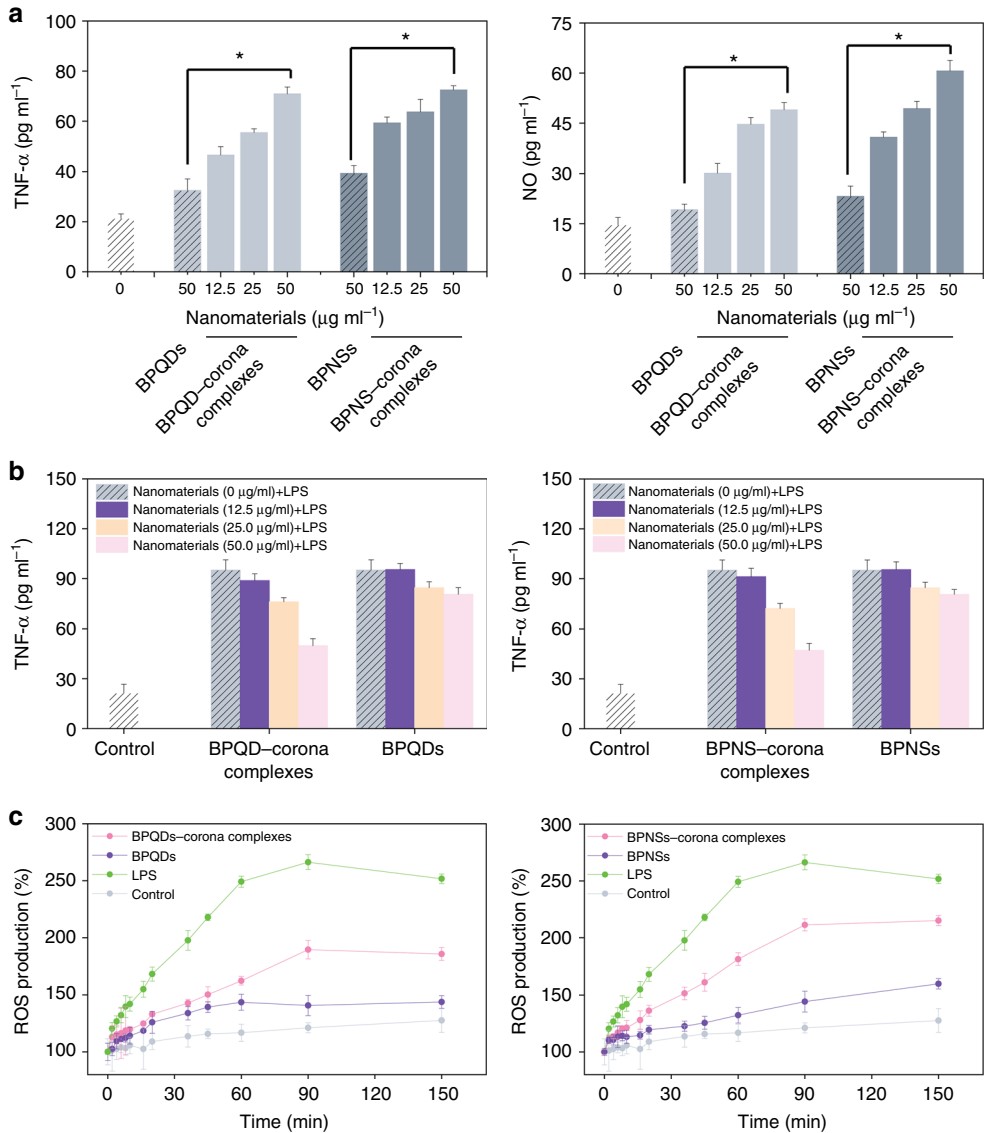

**Fig. 6** Immune perturbation by BP–corona complexes in macrophages. **a** NO and TNF-α generation in dTHP-1 cells. Cells were treated with 50 µg ml⁻¹ BP nanomaterials and an increasing concentration of corona complexes (12.5, 25 and 50 µg ml⁻¹) for 6 h. Values are expressed as the means ± SDs of triplicates. Statistical significance is assessed by Student's *t* test. *$p < 0.05$, **$p < 0.01$. **b** TNF-α secretion in dTHP-1 cells. LPS (20 ng ml⁻¹) was added to the macrophages, which had been pretreated with BP nanomaterials or BP–corona complexes for 6 h. Values are expressed as the means ± SDs of triplicates. **c** ROS overproduction in macrophages exposed to 50 µg ml⁻¹ BP nanomaterials or BP—corona complexes for 6 h. LPS (20 ng ml⁻¹) was set as the positive control. Values are expressed as the means ± SDs of triplicates

systems, including polymer NPs, gold NPs, superparamagnetic iron oxide NPs, etc., have been reported to induce proinflammatory cytokines in immune cells[21,25–28]. To test whether the protein corona bound to BP nanomaterials could increase the secretion of cytokines, macrophage-like cells were treated with either BP nanomaterials or BP–corona complexes. Subsequently, the culture supernatants were collected, and the amount of inflammatory cytokines was detected by enzyme-linked immunosorbent assay (ELISA). Significant changes in cytokine secretion were observed in macrophage cells (dTHP-1 cells and SC cells) treated with corona complexes. As shown in Fig. 5a, BP–corona complexes induced an obvious increase in four inflammatory cytokines: interleukin (IL)−1β, IL-6, IL-8, and interferon (IFN)-γ. In particular, these cytokine secretions in dTHP-1 cells were increased 2–4-fold above background level. However, compared with BP–corona complexes, BP nanomaterials had no significant effect on the secretion of cytokines. IL-9 and IL-10 expression was also not obviously changed upon exposure to BP nanomaterials or BP–corona complexes. Moreover, a similar trend in cytokine expression was observed in macrophages from peripheral blood (SC cells). The secretion of IL-1β, IL-6, IL-8, IL-9 and IL-10 in SC cells was also enhanced by BP–corona complexes (Fig. 5b). However, IFN-γ expression was not significantly changed upon exposure to BP nanomaterials or corona complexes. Taken together, the higher secretion of inflammatory cytokines observed in different human macrophage cells are in support of the proinflammatory effect of BP nanomaterials in the presence of a plasma corona.

Generally, the nuclear factor (NF)-κB pathway has been implicated as a key factor in inflammatory effects induced by foreign materials[29]. The activation of NF-κB results in lκB degradation and nuclear translocation of the p65 dimer[30]. After treatment with BP nanomaterials or corona complexes, the cytoplasmic and nuclear extracts of dTHP-1 cells were analysed by western blot (WB). The results in Supplementary Figure 11 show lκB degradation and nuclear translocation of p65. These changes elicited by BP nanomaterials were much lower than those observed with corona complexes. To further understand the relationship between immune relevant proteins on BP nanomaterials and immune effects, we carefully searched the reported literatures for the effect of every immune protein identified by GO analysis and speculated that BP–corona complexes have more potential immune responses (Supplementary Table 2, 3).

**Immune perturbation of BP–corona complexes in macrophages**. According to previous reports, immunotoxicity of nanomaterials means the effect of nanomaterials on immune cell functions, including inflammatory cytokines secretions, reactive oxygen species (ROS) overexpression, phagocytic activity, proliferative activity, etc[31–33]. Evaluation in vitro of the immunotoxicity of nanomaterials has been reported[34,35]. In addition, the induction of nitric oxide (NO) and tumour necrosis factor (TNF)-α production are molecular markers for the immunotoxicity of nanomaterials[36,37]. To elucidate the immune perturbation in macrophages, we first studied the induction effect of BP–corona complexes on NO and TNF-α. As shown in Fig. 6a, TNF-α and NO production were provoked by BP nanomaterials and corona complexes. The increase in TNF-α and NO expression induced by corona complexes was two times greater than that of BP nanomaterials. The bacterial endotoxin lipopolysaccharide (LPS) is known to induce the release of inflammatory factors in macrophages[25,38,39]. To analyse the effect of corona complexes on macrophage function, we further evaluated stress in macrophage-like cells caused by foreign materials. As the results of Fig. 6b show, after pretreatment with BP–corona complexes, the release

of TNF-α in dTHP-1 cells treated with LPS was significantly inhibited. For example, after dTHP-1 cells were pretreated with 50 μg ml$^{-1}$ BPQD–corona complexes and then incubated with 20 ng ml$^{-1}$ LPS, the secretion of TNF-α was decreased to 49.83 from 95.16 pg ml$^{-1}$ (without pretreatment with BPQD–corona complexes). These results supported the immunotoxicity of BP–corona complexes to macrophages in vitro.

ROS are an important part of the primary immune defence in macrophages against foreign materials. In addition, the expression of ROS has also been recognized as a major toxicological paradigm of nanomaterials[40,41]. The ROS level induced by BP–corona complexes was tested by dichlorofluorescein (DCF) fluorescence assay (Fig. 6c). ROS production was significantly increased in a time-dependent manner in macrophage-like cells treated with BP nanomaterials or BP–corona complexes. Compared with BP nanomaterials, corona complexes induced much higher ROS production. For example, BPQD– and BPNS–corona complexes triggered ROS overproduction in dTHP-1 cells at approximately 185.8% and 215.2% at 150 min, which was much higher than that elicited by BPQDs (143.7%) and BPQNs (159.9%). These findings demonstrated that BP nanomaterials exhibited enhanced immune perturbations in macrophages because of the formation of plasma coronas.

## Discussion

So far, different roles of nanomaterial–protein corona have been defined, including reduction of toxicity to cells, loss of targeting capabilities, extension of blood circulation time, alterations in organ distribution and immunostimulatory and immunosuppressive effects[42–47]. As BP-based nanomedicine is of increasing importance, immunotoxicological assessments of BP nanomaterials are urgently required. Hence, we carried out this study to provide a detailed investigation of the plasma protein corona around BP nanomaterials and its impact on macrophages. Our results show that the size of 2D BP nanomaterials can influence the identity and quantity of plasma proteins that form the corona, and in return, the corona could define the shape of ultra-small 2D nanomaterial–corona complexes but does not change the shape of large 2D nanomaterials. Owing to the enrichment of immune proteins and other opsonins on the surface of BP nanomaterials, the uptake efficiency of BP–corona complexes by macrophages is higher, and the size-dependent cellular uptake pattern of BP–corona complexes is different from that of native nanomaterials. Our data also highlight the importance of the protein corona to the immune perturbation effect on macrophages. As cellular uptake of BP–corona complexes is enhanced, BP nanomaterials bound to opsonins have a distinct impact on the proinflammatory and immune perturbation effect in macrophages. This reveals the pivotal role of the corona in potential innate immune and inflammatory diseases. To further develop BP nanomaterials for effective biological applications, more studies must aim to modify BP nanomaterials to control the formation of the protein corona and its subsequent influence on compatibility for better therapy.

## Methods

**Material and cell lines**. BP crystals were purchased from HWRK Chemical Co., Ltd. (Beijing, China) and stored in a N$_2$ glovebox. N-methyl pyrrolidone (NMP) was purchased from Aladdin Chemistry Co., Ltd (Shanghai, China). HSA-FITC was purchased from Biosynthesis Biotechnology Co., Ltd (Beijing, China). Human lung cancer cells (H1299), human normal hepatic cells (L0-2), human embryonic kidney cells (293T), human monocytic leukaemia cells (THP-1) and human macrophages from peripheral blood (SC) were purchased from American Type Culture Collection (Manassas, VA). All cell lines were not listed by International Cell Line Authentication Committee as cross-contaminated or misidentified (v8.0, 2016). All of the cell lines were authenticated by STR typing and confirmed to be mycoplasma-free by KeyGEN BioTECH Co., Ltd (Nanjing, China). All cells were

cultured in 1640 or Dulbecco's modified Eagle's medium (DMEM) media (Gibco) with 10% FBS (Gibco), 100 units ml$^{-1}$ penicillin, and 50 units ml$^{-1}$ streptomycin at 37 °C in a CO$_2$ incubator (95% relative humidity, 5% CO$_2$). Macrophage-like THP-1 (dTHP-1) cells were differentiated from THP-1 cells by treatment with 100 µg ml$^{-1}$ phorbol myristate acetate for 48 h. All reagents were purchased from Sigma unless otherwise indicated.

**Human plasma.** Blood was taken from ten different apparently healthy donors (according to the Declaration of Helsinki) who had provided informed consent for blood collection and subsequent analysis and stored in tubes containing EDTA to prevent blood clotting. The blood samples were labelled anonymously and could not be traced back to a specific donor. The samples were centrifuged for 5 min at 1700 × $g$ to pellet red and white blood cells. The plasma supernatant was pooled and stored at −80 °C. This plasma was used exclusively throughout the study.

**Preparation of BPNSs and BPQDs.** BPNSs and BPQDs were synthesized according to the previous reports with some slight changes[5,7]. Briefly, 25 mg of the BP power was added to 25 ml of NMP and sonicated in ice bath for 10 h at 600 W. Half of the resulting brown suspension was centrifuged at 1700 × $g$ for 8 min to remove the residual unexfoliated particles, and the supernatant containing BPNSs was collected for further use. The other half of resulting brown suspension was sonicated in an ultrasonic bath continuously for 15 h at 150 W. The dispersed mixture was kept at 0 °C. The resulting sample solution was centrifuged for 20 min at 5200 × $g$, and the supernatant containing BPQDs was gently decanted. Next, the BPQD products could be collected by centrifugation, and the precipitate was repeatedly rinsed with water and resuspended in an aqueous solution.

**Protein–nanomaterial interaction experiments.** BPQDs and BPNSs were incubated with different human plasma concentrations (5, 10, 20, 40, 80 and 100%) at 4 °C for 4 h, and plasma solutions were diluted with phosphate-buffered saline (PBS) keeping the final nanomaterial concentration constant. To obtain protein corona complexes, the mixed solutions were centrifuged to pellet particle–protein complexes. The precipitate was then resuspended three times with 500 µl PBS and centrifuged again for 20 min at 15,000 × $g$ at 4 °C.

**Characterization of nanomaterials.** BPNSs, BPQDs and BP nanomaterial–protein corona complex solutions were dispersed onto holey carbon film on copper grids and then observed by TEM on a JEOL JEM-1200EX at an acceleration voltage of 120 kV. EDX was conducted on the TEM.

**Zeta potential and DLS measurement.** BP nanomaterials (BPQDs and BPNSs) were incubated with plasma at 4 °C for 4 h. The products were washed three times with 500 µl PBS to thoroughly remove unbound plasma protein and centrifuged again for 20 min at 15,000 × $g$ at 4 °C to collect BP–corona complexes. After resuspension in PBS, the zeta potential and size of BP–corona complexes were measured on a Nano-ZS instrument (Malvern Instruments Limited) and DLS (Brookhaven BI-200SM). In addition, the zeta potential and size of BPQDs and BPNSs were also measured in PBS. All experiments were conducted at least twice to ensure the reproducibility of the results.

**SDS-polyacrylamide gel electrophoresis.** SDS-PAGE was carried out according to standard procedures[48]. Corona complexes were separated by 12% SDS-PAGE using the SDS-PAGE Preparation Kit (Sangon Biotech Co., Ltd, Shanghai, China). Then proteins were visualized by staining with Coomassie brillian blue R-250. All experiments were conducted at least twice to ensure the reproducibility of the results.

**Protein identification and classification.** The BP–corona complexes were collected by the above methods. Then one volume of SDT buffer (4% SDS, 100 mM dithiothreitol (DTT), 150 mM Tris-HCl pH 8.0) was added, and the solution was boiled for 15 min and centrifuged at 14,000 × $g$ for 20 min. Digestion of protein (200 µg for each sample) was performed according to the FASP procedure described by Mann et al[49]. In all, 200 µl UA buffer (8 M Urea, 150 mM Tris-HCl, pH 8.0) were added to removed the detergent, DTT and other low-molecular components by ultrafiltration (Microcon units, 30 KD). Reduced cysteine residues were blocked with 100 µl iodoacetamide (0.05 M in UA buffer) and the samples were incubated for 20 min in the dark. The filter was washed with 100 µl UA buffer for three times and then with 100 µl NH$_4$HCO$_3$ (25 mM) twice. Trypsin (3 µg, Promega) in 40 µl NH$_4$HCO$_3$ (25 mM) was added to digest the proteins at 37 °C for 12 h, and the resulting peptides suspension was collected by filtration. The content of peptide was determined with ultraviolet light spectral density at 280 nm using an extinction coefficient of 1.1 of 0.1% solution. Each fraction was injected into a Q-Exactive mass spectrometer (Thermo Scientific) for LC-MS/MS analysis. The MS/MS spectra were searched with MASCOT engine (Matrix Science, London, UK; version 2.4). The following option was used: peptide mass tolerance = 20 ppm, MS/MS tolerance = 0.1 Da, enzyme = Trypsin, missed cleavage = 2, fixed modification: carbamidomethyl (C), and variable modification: oxidation (M)[50–52]. The identified proteins were retrieved from the UniProtKB human database (Release 2017

_02) in FASTA format. In this study, the top 10 blast hits with $E$-values <1e-3 for each query protein were retrieved and loaded into Blast2GO (Version 3.3.5) for GO mapping and annotation. The GO analysis was carried out with Blast2GO[53]. All experiments were conducted at least twice to ensure the reproducibility of the results.

**Cytotoxicity assay.** The cytotoxicities of BP nanomaterials and BP–corona complexes to cells were determined by MTT assay. Cells (2 × 10$^4$ cells per well) were seeded in 96-well tissue culture plates for 24 h and then incubated with 100 µl of fresh medium (0% FBS) containing different concentrations of nanomaterials (100, 50, 25, 12.5 µg ml$^{-1}$) for 48 h. Subsequently, 30 µl per well of MTT solution (KeyGEN BioTECH Co., Ltd, Nanjing, China) was added and incubated for 4 h. After incubation, the culture medium was removed and replaced with 150 µl per well dimethyl sulphoxide. The colour intensity of the medium was measure at 550 nm using a microplate reader (Tecan Infinite M1000 PRO) to calculate the cell viability. In addition, we also analysed the cytotoxicities of BP nanomaterials and BP–corona complexes in 1640 or DMEM media (10% FBS) following the methods above. All experiments were conducted at least twice to ensure the reproducibility of the results.

**Cellular uptake of BP nanomaterials and corona complexes.** Macrophage-like dTHP-1 cells (1.2 × 10$^6$ cells ml$^{-1}$) were seeded in 6-well tissue culture plates and grown to ~90% confluence. BPQDs and BPNSs (200 µg) were incubated with 200 µl of 100% plasma for 4 h at 4 °C. Corona complexes were obtained as described above and resuspended in 1 ml 1640 media (0% FBS) to prepare for cell culture. Meanwhile, 200 µg of free BPQDs and BPNSs were resuspended in 1 ml of DMEM (0% FBS). Growth media in plates was replaced with media containing nanomaterials and corona complexes. Cells were incubated with nanomaterials for 1, 3 or 6 h at 37 °C. Finally, the medium was removed from the wells, and the cells were washed three times with PBS to clear away nanomaterials outside of the cells. Next, 200 µl of 1% Triton X-100 in 0.1 M NaOH solution was added to lyse the cells. After treatment, P content in the lysates was determined using ICP-MS. In addition, we also measured the cellular uptake efficiency of BP–corona complexes in 1640 media (10% FBS) following the methods above. All experiments were conducted at least twice to ensure the reproducibility of the results.

**Preparation of fluorescent BP–corona complexes and stability analysis.** BPQDs and BPNSs (200 µg) were incubated with 200 µl of 100% plasma (mixed with 10 µg HSA-FITC) for 4 h at 4 °C. To obtain protein corona complexes, the mixed solution was centrifuged to pellet the corona complexes. The precipitate was then resuspended three times in 500 µl PBS and centrifuged again for 20 min at 15,000 × $g$ at 4 °C. The green fluorescence intensity of the BPNS–protein corona complexes was tested by SDS-PAGE.

The stability of fluorescent BP–corona complexes was determined according to previous literature[36]. Briefly, fluorescent BP–corona complexes were suspended in the dTHP-1 cells lysate with shaking at 37 °C. At specific times during incubation, the fluorescence intensity of the supernatant was measured by microplate reader (Tecan Infinite M1000 PRO) with the excitation and emission wavelengths set to 490 and 520 nm, respectively, after centrifugation to remove the corona complexes. The fluorescence intensity of the corona complexes was also detected as the control. All experiments were conducted at least twice to ensure the reproducibility of the results.

**Intracellular localization of corona complexes.** For fluorescence microscopy, dTHP-1 cells were first seeded onto 2-cm culture dishes at a density of 8.0 × 10$^4$ cells ml$^{-1}$ for 24 h. Fluorescent BP–corona complexes (200 µg) were obtained following the methods above and then resuspended in 1 ml DMEM (0% FBS) to prepare for cell culture. Cells were incubated with nanomaterials for 6 h at 37 °C, and then the medium was removed from the dish. After rinsing with PBS three times to remove residual nanomaterials, cells were immobilized with 1 ml 4% paraformaldehyde for 15 min. Then the cell nucleus and lysosome were stained with 4,6-diamidino-2-phenylindole and Lyso Tracker, respectively (Beyotime Biotechnology Co., Ltd, Shanghai, China); the cells were monitored using a fluorescence microscope (ZEISS).

For TEM, dTHP-1 cells were cultivated in 6-well plates at 1.0 × 10$^6$ cells per well for 24 h. BPQDs and BPNSs (200 µg) were incubated with 200 µl of 100% plasma for 4 h at 4 °C. Corona complexes were obtained as described above and resuspended in 1 ml 1640 media (0% FBS) to prepare for cell culture. Meanwhile, 200 µg of free BPQDs and BPNSs were resuspended in 1 ml DMEM (0% FBS). Cells were incubated with nanomaterials for 6 h at 37 °C, and the medium was removed from the dish. After rinsing with PBS three times to remove the residual nanomaterials, cells were transferred to fixative, stained, sliced and imaged by TEM (Hitachi HT7700) according to previous literature[19]. All experiments were conducted at least twice to ensure the reproducibility of the results.

**Cytokine secretion assay.** To analyse the cytokine secretion by different macrophage-like cells, the cells were seeded onto 96-well plates at 1.0 × 10$^5$ cells per well for 24 h. BPNSs, BPQDs and corona complexes were resuspended in culture media (0% FBS), added to each well at different concentrations (0, 12.5, 25, 50 µg

ml$^{-1}$) and incubated for 6 h. Then the supernatant was collected and the cytokines (IL-1β, IL-6, IL-8, IL-9, IL-10, IFN-γ and TNF-α) and NO were measured using ELISA kits (R&D Systems, CA, USA).

To analyse the immune perturbation effect of corona complexes and BP nanomaterials, dTHP-1 cells were seeded onto 96-well plates at $1.0 \times 10^5$ cells per well for 24 h. Nanomaterials were resuspended in serum-free media, added into each well at different concentrations (0, 12.5, 25, 50 µg ml$^{-1}$) and incubated for 6 h. Then the supernatant was removed, and the cells were washed with PBS three times. Subsequently, macrophages were further treated with LPS (20 ng ml$^{-1}$) for 6 h. Finally, the supernatants were collected, and the cytokines were determined with ELISA kits. All experiments were conducted at least twice to ensure the reproducibility of the results.

**Measurement of ROS in macrophages.** The effect of BP nanomaterials and corona complexes on intracellular ROS generation in macrophage cells was detected with a Reactive Oxygen Species Assay Kit (KeyGEN BioTECH Co., Ltd, Nanjing, China) following the manufacturer's protocol. Briefly, dTHP-1 cells were seeded onto 96-well plates at $1.0 \times 10^6$ cells per well for 24 h, and then the culture medium was replaced with 100 µl of DCF-DA/DMEM44 medium (final DCF concentration is 10 µM) and incubated at 37 °C for 20 min. Then BP nanomaterials (100 µg ml$^{-1}$) and corona complexes (100 µg ml$^{-1}$) were added to the wells. The fluorescence intensity of DCF was measured by microplate reader (Tecan Infinite M1000 PRO) with the excitation and emission wavelengths at 488 and 525 nm, respectively, and the fluorescence intensity was used to evaluate the ROS level in macrophage cells. All experiments were conducted at least twice to ensure the reproducibility of the results.

**Western blot analysis.** Macrophage-like dTHP-1 cells were treated with BP nanomaterials (100 µg ml$^{-1}$) and BP–corona complexes for 6 h. Then the cells were washed with PBS three times and collected. The cytoplasmic and nuclear extracts were prepared using the Minute™ Cytoplasmic and Nuclear Fractionation Kit (Invent Biotechnologies, Inc), and the protein concentration was determined by the Super-Bradford Protein Assay Kit (CWBiotech, Inc, Beijing, China). The expression levels of lκB in cytoplasm extracts and p65 in nuclear extracts were determined by WB according to previous literature[54]. Briefly, the extracts were first separated by 12% SDS-PAGE and transferred to a polyvinylidene difluoride membrane (Bio-Rad, CA, USA). The membrane was blocked with 5% BSA in PBS at 25 °C for 2 h and then incubated with antibodies at 4 °C overnight. The expression of β-actin was used as the internal standard. The uncropped WB images were included in Supplementary Figure 12. The sources of primary antibody against the following protein were: β-actin (A1978, 1:10,000 for WB) from sigma; p65 in nuclear (ab16502, 1:1000 for WB), and lκB (15095, 1:1000 for WB) from Abcam. The appropriate secondary antibodies (1:10,000 for WB) were purchased from Abcam.

**Statistical analysis.** All the data are expressed as mean ± standard deviation. Differences between different experimental groups were analysed by two-tailed Student's $t$ test. One-way analysis of variance was used in multiple group comparisons. Statistical analysis was performed using SPSS statistical program version 13 (SPSS Inc., Chicago, IL). Differences with $p < 0.05$ (*) or $p < 0.01$ (**) were considered statistically significant.

**Data availability.** All data are available from the corresponding authors upon reasonable request. The LC-MS/MS data have been deposited to the ProteomeXchange Consortium via the PRIDE[55] partner repository with the data set identifier PXD009645.

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

## Acknowledgements

This work was written in memory of Dr. Qing Huang for his great inspiration for this research. Financial support was provided by the National Science Foundation of China (21622103, 21571098 and 21671099), the Natural Science Foundation of Jiangsu Province (BK20160022) and Shenzhen Basic Research Program (JCYJ20170413150538897 and GJHS20170310093035206).

## Author contributions

J.M and Q.X. carried out the experimental works; J.M. and W.W. wrote the manuscript; J.Z. and W.W. guided the project.

## Additional information

**Competing interests:** The authors declare no competing interests.

