## [Peer Review File · Nature Communications]

Reviewers' Comments:

Reviewer #1:

Remarks to the Author:

Mo et al. studied the influence of the adsorbed protein corona on the in vitro response of macrophages to black phosphorus nanoparticles (BPNPs). This is an important issue both from the perspective of safety – ie the immune response to these nanoparticles upon intentional or accidental exposure – as well as from perspective of applications in the emerging field of nano-immunotherapy – ie the development of vaccines and immunologically active nanomaterials. While the authors have limited their study to BPNPs – a relatively niche material – the structure of the study and general findings will be of interest to a wide audience of nanomaterials researchers.

Overall, the design and execution of the study is good. However, there are several key issues that have not been adequately addressed. As a result, in its current form, I cannot recommend the manuscript for publication in Nature Communications. However, I have included a list of suggested modifications below. If these issues are adequately addressed, I believe the manuscript will be suitable for publication.

UNADDRESSED ISSUES:

- A key finding is the influence of adsorbed plasma proteins on the cytokine expression of RAW264.7 macrophage-like cells. However, the plasma is of human origin and the cells are of murine origin. How do species-specific effects contribute to the observed results? Two questions should be addressed experimentally: 1) Do similar results hold if Raw264.7 cells are exposed to BPNPs incubated in mouse plasma? 2) Do similar results hold if BPNPs are incubated in human plasma and then exposed to human macrophage-like cells (for example differentiated THP-1 cells).
- Related to the above point, to what extent are the observed results specific to the cell type used as a model for macrophages? In other words, if a different murine macrophage-like cell line were used, would a similar trend in cytokine expression be observed? This question could be interrogated experimentally by studying cytokine secretion from a different macrophage-like cell line (for e.g. J774.1 cells).
- The authors find a significant increase in the release of cytokines from RAW264.7 cells when exposed to BPNPs after incubation in plasma. However, they do not adequately discuss the relevance of these findings to potential immunotoxicity and/or immunotherapeutic applications of these materials. A more thorough discussion is warranted.
- While the authors observe a striking increase in cytokine secretion upon exposure to BPNPs incubated in plasma vs those not incubated in plasma, there is inadequate discussion of how the specific adsorbed plasma proteins identified via LC-MS/MS may be contributing to the observed effect.
- The authors report a significant size increase in BPODs after plasma incubation by TEM analysis. However, a proper counting experiment was not performed to verify the statistical significance of this claim. The authors should therefore assess a significant number of particles (in an unbiased way) for both the as-synthesized and plasma-incubated BPODs from TEM fields to assess the difference in average diameter between the conditions. In addition, drying artefacts and the inherently low electron density of biological material may bias TEM observations. This possibility should be discussed and, if appropriate, the authors should also use dynamic light scattering (or equivalent technique) to evaluate the 'wet' diameter of the samples both before and after incubation with plasma.
- The aggregation state of the BPNPs used in this study may provide an additional confounding variable influencing the observed cell results. Using an appropriate methodology (TEM or DLS for example), authors should assess the level of aggregation of their materials both after plasma exposure and in cell culture media. Using TEM analysis, the authors should be able to assess the number of BPNPs per aggregate cluster. A distribution of this quantity should be reported.
- Authors state that "BP nanosheets (BPNs) were able to bind 25.0 percent of immune proteins from plasma, while BP quantum dots (BPODs) bound 42.4 percent of immune proteins" since the

absolute number of immune proteins in blood plasma is unknown, it would be more appropriate to state the total number of immune proteins identified, not the fraction.

- For zeta potential measurements, were the particles first separated from unbound protein? The unbound plasma proteins may interfere with light scattering measurements, particularly for small nanoparticles in the <20nm range, as is the case for BPQDs here. This should be made clear in the materials & methods section.
- In the methods section, plasma was isolated from different donors. Was this plasma then pooled? Since the donor is known to influence the composition of plasma, comparing samples incubated in plasma from different donors could lead to donor-dependent effects on the protein corona. This should be indicated explicitly in the manuscript.
- Authors suggest that differences in the composition of the protein corona between BPNS and BPODs is due to nanoparticle size. However, such differences may also result from differences in the surface chemistry introduced during the manufacturing process. Would the authors make a reasoned argument as to why the observed effects are due to size and not surface chemistry?
- Authors track intracellular localization of nanoparticle-protein complexes using a FITC-labeled BSA adsorbed to the surface. However, upon internalization (or even in the surrounding cell culture media), it is possible that the FITC-labeled BSA might desorb or be competitively displaced by protein in the surrounding media. In which case, in the fluorescence microscopy studies, the authors are simply observing free BSA and not BSA bound to nanoparticles. Additional studies should be undertaken to assess the stability of BSA-nanoparticle complexes in both plasma and artificial lysosomal fluid to verify that BSA remains attached to the particles under realistic physiological conditions.
- Authors carried out cell assays under serum-free conditions – which is not standard culture practice. Additional cell studies should be performed to study the effect of serum on cell uptake and serum-free conditions. In this case, nanoparticles could be pre-incubated with plasma and resuspended in either DMEM + 0% FBS or DMEM + 10% FBS and the effects compared.
- The relevance of findings to other nanomaterial systems should be discussed.
- Some typographical errors. Thorough proof-reading and copy-editing needed.

Reviewer #2:

Remarks to the Author:

In their manuscript entitled "Revealing the Immune Response of Black Phosphorus Nanomaterials by Understanding the Protein Corona" Mo et al. investigate the role of the protein corona on the physicochemical properties and biocompatibility of black phosphorous nanoparticles. (BPN) While the protein corona of BPNs has not yet been investigated, there are currently several methodological, experimental and conceptual limitations that make it impossible to judge the validity of the presented data and preclude publication at this point. Data seem to be preliminary and insight into the current state of the art in nanoparticle proteomics and characterization of nanoparticle biology seems to be lacking.

Major points:

1. What do the authors refer to by "the immune response of nanoparticles" - Many statements do not make any sense from an immunological perspective, e.g. Line 123 "Immune Response of BP Nanomaterials-corona Complexes" – do the authors refer to an immune response of the host against the nanomaterials? Investigating uptake of BPN and the cytokine secretion in serum free media of a single macrophage-like cell line is certainly not sufficient to characterize the host immune response. Any in vivo data are lacking.
2. The proteomics data are insufficiently described. No data are provided except on a highly aggregated level. The entire dataset is not presented (neither in the manuscript nor supplementary information), making it impossible to judge the validity of results. No information is provided regarding number of biological or technical replicates for the proteomic analyses. The authors must provide the entire datasets in the supplementary information, data should include (at

least): Protein Identifier, Scores, FDR, number of peptides per proteins, quantification values from all biological and technical replicates. Methods for proteomic analysis are insufficiently described. No details regarding lysis buffers or LC conditions are provided. No details for search parameters are provided. Please refer to MCP reporting guidelines for proper reporting of proteomic datasets. Additionally, I highly recommend submitting the datasets (rawdata and search results) to a public repository (i.e. ProteomeXchange) to allow other researchers to access the data.

3. The authors interpretations of the proteomics data are strange. The authors only look at numbers of proteins and actual quantification results are hidden from the reader. What is actually meant by "discovered that 42.4 percent of immune proteins from plasma were attracted to BPQDs and 25 percent of immune proteins were attracted to BPNSs".

4. Most recent contributions by other groups concerning the role of the protein corona on nanoparticle biology have been ignored by the authors.

5. Incubation of nanoparticles with serum/plasma at 4°C is a highly artificial setting – why were the experiments not performed at 37°C?

6. The cytokine secretion assay was performed in serum-free media, which typically leads to starvation and subsequent extensive cell death.

The manuscript contains numerous typographical errors and should be proofread by a native speaker.

Responses to Referee's Comments

Reviewer 1

Mo et al. studied the influence of the adsorbed protein corona on the in vitro response of macrophages to black phosphorus nanoparticles (BPNPs). This is an important issue both from the perspective of safety – ie the immune response to these nanoparticles upon intentional or accidental exposure – as well as from perspective of applications in the emerging field of nano-immunotherapy – ie the development of vaccines and immunologically active nanomaterials. While the authors have limited their study to BPNPs – a relatively niche material – the structure of the study and general findings will be of interest to a wide audience of nanomaterials researchers.

Overall, the design and execution of the study is good. However, there are several key issues that have not been adequately addressed. As a result, in its current form, I cannot recommend the manuscript for publication in Nature Communications. However, I have included a list of suggested modifications below. If these issues are adequately addressed, I believe the manuscript will be suitable for publication.

Query 1

(1) A key finding is the influence of adsorbed plasma proteins on the cytokine expression of RAW264.7 macrophage-like cells. However, the plasma is of human origin and the cells are of murine origin. How do species-specific effects contribute to the observed results? Two questions should be addressed experimentally: 1) Do similar results hold if Raw264.7 cells are exposed to BPNPs incubated in mouse plasma? 2) Do similar results hold if BPNPs are incubated in human plasma and then exposed to human macrophage-like cells (for example differentiated THP-1 cells).

Response

We thank the reviewer for this kind advice. Following the above two points, we carefully deliberated on the species-specific effects and believed that it would be more exact to examine the effect of corona complexes on cytokine secretion in human macrophage cells. We measured the cytokine level in human macrophage-like dTHP-1 cells. The results showed that BP-corona complexes induced an obvious increase in 4

inflammatory cytokines: interleukin (IL)-1 β , IL-6, IL-8 and interferon (IFN)- γ . In particular, these cytokine secretions in dTHP-1 cells were increased 2-4-fold above background level. However, compared with BP-corona complexes, BP nanomaterials had no significant effect on these cytokine secretions. IL-9 and IL-10 were also expressed without a significant change upon exposure to BP nanomaterials or BP-corona complexes (Figure 4a).

Revisions Made

(Please refer to page 11, line 193-200).

Subsequently, the culture supernatants were collected, and the amount of inflammatory cytokines was detected by enzyme-linked immunosorbent assay (ELISA). Significant changes in cytokine secretion were observed in macrophage cells (dTHP-1 cells and SC cells) treated with corona complexes. As shown in Figure 4a, BP-corona complexes induced an obvious increase in 4 inflammatory cytokines: interleukin (IL)-1 β , IL-6, IL-8, and interferon (IFN)- γ . In particular, these cytokine secretions in dTHP-1 cells were increased 2-4-fold above background level. However, compared with BP-corona complexes, BP nanomaterials had no significant effect on the secretion of cytokines. IL-9 and IL-10 expression was also not obviously changed upon exposure to BP nanomaterials or BP-corona complexes.

Figure 4a. Cytokine secretion of macrophage-like dTHP-1 cells. Cells were treated with 50 $\mu\text{g/ml}$ BP nanomaterials and an increasing concentration of corona complexes (12.5, 25 and 50 $\mu\text{g/ml}$) for 6 h. Values are expressed as the means \pm SDs of triplicates. * $p<0.05$, ** $p<0.01$

Query 2

(2) Related to the above point, to what extent are the observed results specific to the cell type used as a model for macrophages? In other words, if a different murine macrophage-like cell line were used, would a similar trend in cytokine expression be observed? This question could be interrogated experimentally by studying cytokine secretion from a different macrophage-like cell line (for e.g. J774.1 cells).

Response

We thank the reviewer for this kind advice. Following the above two points, we further analysed the effect of BP-corona complexes on cytokine secretion in another macrophage cell line (macrophage SC cell). The results showed a similar trend in cytokine expression in dTHP-1 cells and SC cells. Compared with BP nanomaterials, BP-corona complexes induced a significant ($p<0.05$) increase in 5 cytokine levels: interleukin (IL)-1 β , IL-6, IL-8, IL-9 and IL-10 (Figure 4b). Interferon (IFN)- γ was

expressed without a significant change upon exposure to BP nanomaterials or corona complexes.

Revisions Made

(Please refer to page 11, line 200-205).

Moreover, a similar trend in cytokine expression was observed in macrophages from peripheral blood (SC cells). The secretion of IL-1 β , IL-6, IL-8, IL-9 and IL-10 in SC cells was also enhanced by BP-corona complexes (Figure 4b). However, IFN- γ expression was not significantly changed upon exposure to BP nanomaterials or corona complexes. Taken together, the higher secretion of inflammatory cytokines observed in different human macrophage cells are in support of the proinflammatory effect of BP nanomaterials in the presence of a plasma corona.

Figure 4b. Cytokine secretion of SC cells. Cells were treated with 50 μ g/ml BP nanomaterials and an increasing concentration of corona complexes (12.5, 25 and 50 μ g/ml) for 6 h. Values are expressed as the means \pm SDs of triplicates. * p <0.05, ** p <0.01

(3) The authors find a significant increase in the release of cytokines from RAW264.7 cells when exposed to BPNPs after incubation in plasma. However, they do not adequately discuss the relevance of these findings to potential immunotoxicity and/or immunotherapeutic applications of these materials. A more thorough discussion is warranted.

Response

We thank the reviewer for the good question. Following this suggestion, we further analysed the potential immunotoxicity of corona complexes. According to literatures, NO and TNF- α production were suggested to be molecular markers for immunotoxicity of nanomaterials (Old. *Science*. 230, 630-632 (1985); Dobrovolskaia *et al. Nat. Nanotechnol.* 2, 469-478 (2007)). Therefore, we evaluated the generation of NO and TNF- α in dTHP-1 cells to elucidate the perturbation of corona complexes on macrophages. The results showed a promoting effect of corona complexes on the generation of NO and TNF- α (Figure 5a). In addition, ROS production is a part of primary immune defence in macrophage cell against foreign materials (Yan *et al. ACS nano.* 5, 4581-4591 (2011)). To determine the intracellular ROS, we have monitored the increase of fluorescence intensity of dichlorofluorescein (DCF) in dTHP-1 cells. As shown in Figure 5c, the increased intracellular ROS generated by BP-corona complexes was sharper than that by BP nanomaterials. For example, BPQD- and BPNS-corona complexes triggered ROS overproduction in dTHP-1 cells at about 185.8% and 215.2% at 150 min, respectively, which was much higher than that elicited by BPQDs (143.7%) and BPQNs (159.9%). Finally, we studied the dTHP-1 cells pretreated with corona complexes responded to bacterial endotoxin LPS, which is known to induce the release of inflammatory factors in macrophages. As Figure 5b shown, the TNF- α production reduced from 95.16 pg/ml (dTHP-1 treated with 20 ng/ml of LPS) to 49.83 pg/ml (dTHP-1 pretreated with 50 μ g/ml BPQD-corona complexes and then treated with 20 ng/ml LPS), indicating that corona complexes disturbed the function of macrophages. A similar perturbation effect of BPNS-corona complexes could be observed. Taken together, the high level of TNF- α , NO and ROS production and low-level stress response to foreign materials are in agreement with the potential immunotoxicity of BP-corona complexes.

Revisions Made

(Please refer to page 14-15, line 229-251).

Immune Perturbation of BP-corona Complexes in Macrophages. Previous reports have shown that the induction of nitric oxide (NO) and tumour necrosis factor (TNF)- α production is a molecular marker for the immunotoxicity of nanomaterials. To elucidate the immune perturbation in macrophages, we first studied the induction effect of BP-corona complexes on NO and TNF- α . As shown in Figure 5a, TNF- α and NO production were provoked by BP nanomaterials and corona complexes. The increase in TNF- α and NO expression induced by corona complexes was 2 times greater than that of BP nanomaterials. The bacterial endotoxin LPS is known to induce the release of inflammatory factors in macrophages. To analyse the effect of corona complexes on macrophages function, we further evaluated stress in macrophages caused by foreign materials. As the results of Figure 5b show, after pretreatment with BP-corona complexes, the release of TNF- α in dTHP-1 cells treated with LPS was significantly inhibited. For example, after dTHP-1 cells were pretreated with 50 μ g/ml BPQD-corona complexes and then incubated with 20 ng/ml LPS, the secretion of TNF- α was decreased to 49.83 pg/ml from 95.16 pg/ml (without pretreatment with BPQD-corona complexes). These results supported the immunotoxicity of BP-corona complexes to macrophages.

Reactive oxygen species (ROS) are an important part of the primary immune defence in macrophages against foreign materials. In addition, the expression of ROS has also been recognized as a major toxicological paradigm of nanomaterials. The ROS level induced by BP-corona complexes was tested by dichlorofluorescein (DCF) fluorescence assay (Figure 5c). ROS production was significantly increased in a time-dependent manner in macrophage cells treated with BP nanomaterials or BP-corona complexes. Compared with BP nanomaterials, corona complexes induced much higher ROS production. For example, BPQD- and BPNS-corona complexes triggered ROS overproduction in dTHP-1 cells at approximately 185.8% and 215.2% at 150 min, which was much higher than that elicited by BPQDs (143.7%) and BPQNs (159.9%). These findings demonstrated that BP nanomaterials exhibited enhanced immune perturbations in macrophages because of the formation of plasma coronas.

Figure 5. Immune perturbations of BP-corona complexes in macrophages. (a) NO and TNF- α generation in dTHP-1 cells. Cells were treated with 50 $\mu\text{g/ml}$ BP nanomaterials and an increasing concentration of corona complexes (12.5, 25 and 50 $\mu\text{g/ml}$) for 6 h. Values are expressed as the means \pm SDs of triplicates. * $p < 0.05$. (b) TNF- α secretion in dTHP-1 cells. LPS (20 ng/ml) was added to the macrophages which had been pretreated with BP nanomaterials or BP-corona complexes for 6 h. Values are expressed as the means \pm SDs of triplicates. (c) ROS overproduction in macrophages exposed to 50 $\mu\text{g/ml}$ BP nanomaterials or BP-corona complexes for 6 h. Values are expressed as the means \pm SDs of triplicates.

Query 4

(4) While the authors observe a striking increase in cytokine secretion upon exposure to BPNPs incubated in plasma vs those not incubated in plasma, there is inadequate discussion of how the specific adsorbed plasma proteins identified via LC-MS/MS may be contributing to the observed effect.

Response

We thank the reviewer for the good question. Following this suggestion, we carefully added more discussions about the relationship between the identified corona proteins and the increases in cytokine secretion.

Generally, NF- κ B pathway has been implicated as a key factor in inflammatory effects induced by foreign materials (Verma *et al. Nat. Rev. immunol.* 2, 975 (2002)). The activation of NF- κ B results in I κ B degradation, and nuclear translocation of the p65 dimer (Ghosh *et al. Oncogene.* 25, 6758 (2006)). After BP nanomaterials or corona complexes treatment, the cytoplasmic and nuclear extracts of dTHP-1 cells were analysed by Western blot. The results in Figure S11 showed I κ B degradation and nuclear translocation of the p65. These changes elicited by BP nanomaterials were much lower than corona complexes. For further understanding of the relationship between immune protein on BP nanomaterials and immune effect, we carefully searched reported literatures about the effect of every immune protein (identified by gene ontology analysis) and speculated that BP-corona complexes have more potential immune response (Table S5).

Revisions Made

(Please refer to page 12-14, line 211-227).

Generally, the NF- κ B pathway has been implicated as a key factor in inflammatory effects induced by foreign materials. The activation of NF- κ B results in I κ B degradation and nuclear translocation of the p65 dimer. After treatment with BP nanomaterials or corona complexes, the cytoplasmic and nuclear extracts of dTHP-1 cells were analysed by Western blot. The results in Figure S11 show I κ B degradation and nuclear translocation of p65. These changes elicited by BP nanomaterials were much lower than those observed with corona complexes. To further understand the relationship between immune proteins on BP nanomaterials and immune effects, we carefully searched the reported literatures for the effect of every immune protein identified by GO analysis and speculated that BP-corona complexes have more potential immune responses (Table S5).

Figure 5. Activation of NF-κB pathways by BP nanomaterials and corona complexes. Macrophage-like dTHP-1 cells were treated with nanomaterials at 100 μg/ml for 6h.

Query 5

(5) The authors report a significant size increase in BPQDs after plasma incubation by TEM analysis. However, a proper counting experiment was not performed to verify the statistical significant of this claim. The authors should therefore assess a significant number of particles (in an unbiased way) for both the as-synthesized and plasma-incubated BPQDs from TEM fields to assess the difference in average diameter between the conditions. In addition, drying artefacts and the inherently low electron density of biological material may bias TEM observations. This possibility should be discussed and, if appropriate, the authors should also use dynamic light scattering (or equivalent technique) to evaluate the ‘wet’ diameter of the samples both before and after incubation with plasma.

Response

We thank the reviewer for this kind advice. Following this suggestion, we have repeated statistical analysis of BPQDs and BPQD-proteins corona complexes (Figure 1b and 1f). According to statistical results, the average lateral size of BPQDs and BPQD-corona complexes is 5.7 ± 1.0 nm and 371 ± 9.1 nm, respectively. Besides, we further evaluate the ‘wet’ diameter of the samples both before and after incubation with plasma using DLS (Figure 1h). The results showed that the size of BPQDs and BPQDs-corona complexes is 5.6 ± 1.4 nm and 362.5 ± 5.6 nm, respectively.

Revisions Made

(Please refer to page 5, line 77-79).

The zeta-potential values of BPQDs decreased from -20.6 mV to -6.4 mV, and the “wet” diameter increased from 5.6 ± 1.4 nm to 362.5 ± 5.6 nm (Figure 1h, 1i and Table S1).

Figure 1. (b) TEM image of BPQDs. Inset: statistical analysis of the size of 100 BPQDs determined by TEM. (f) TEM image of BPQD-corona complexes. Inset: statistical analysis of the size of 100 BPQD-corona complexes determined by TEM. (h) DLS analysis of BPQDs and BPQD-corona complexes.

Query 6

(6) The aggregation state of the BPNPs used in this study may provide an additional confounding variable influencing the observed cell results. Using an appropriate methodology (TEM or DLS for example), authors should assess the level of aggregation of their materials both after plasma exposure and in cell culture media. Using TEM analysis, the authors should be able to assess the number of BPNPs per aggregate cluster. A distribution of this quantity should be reported.

Response

We thank the reviewer for this kind advice. Following this suggestion, we have evaluated the average hydrodynamic diameter of BP nanomaterials and BP-corona complexes in different solution (PBS and serum-free media) through DLS. As shown in

Figure 1g-h, the results were typical of a monomodal distribution of BP nanomaterials and BP-corona complexes, and the corresponding average hydrodynamic diameter is 338.4 ± 2.3 nm (BPNSs), 365.3 ± 5.9 nm (BPNS-corona complexes), 5.6 ± 1.4 nm (BPQDs) and 362.5 ± 5.6 nm (BPQD-corona complexes). The hydrodynamic diameter of BP nanomaterials and BP-corona complexes measured in serum-free media was similar to that in PBS (Table S1).

Besides, we have carefully searched previous literatures about the effect of aggregation state of nanoparticles on cellular assays. As the Figure 1b in Chen's literatures, the silica nanoparticles, which were slightly aggregated, were seen to be monodisperse, and this state had no obvious impact on cell results (*Chen et al. Adv. Funct. Mater.* 24, 2754-2763 (2014)). The dispersion state of BP nanomaterials and BP-corona complexes is similar to other nanomaterials reported in different literature (i.e., *Chu et al. Nat. Commun.* 7, 12967 (2016); *Shi et al. Nat. Commun.* 8, 357 (2017); *Chen et al. Adv. Funct. Mater.* 24, 2754-2763 (2014)). Hence, we believe the slight aggregation state of BPNPs make no obvious impact on cell results.

Figure 1b. TEM images of MSNs. Scale bars is 100 nm. (*Chen et al. Adv. Funct. Mater.* 24, 2754-2763 (2014))

Revisions Made

(Please refer to page 3, line 51-54).

From the TEM image results, BPNSs were found to be free-standing with a lateral size of approximately 300 nm, and BPQDs were found to be approximately 5 nm in size (Figure 1a, b). According to a statistical TEM analysis of 100 BPQDs (Figure 1e), the average lateral size of a BPQD is 5.7 ± 1.0 nm.

(Please refer to page 4-5, line 73-79).

The size of BPNS-corona nanomaterials was also investigated by DLS analysis (Figure 1g and Table S1). The results showed that the average size of BPNSs changed from 338.4 ± 2.3 nm to 365.3 ± 5.9 nm. Unexpectedly, after the protein corona was formed on the surface of BPQDs, BPQDs were redefined from ultra-small nanosheets to spherical BPQD-corona complex nanoparticles, and the nanoparticles appeared to be bulky particles with a diameter of approximately 371.9 ± 9.1 nm (Figure 1b, f). The zeta-potential values of BPQDs decreased from -20.6 mV to -6.4 mV, and the “wet” diameter increased from 5.6 ± 1.4 nm to 362.5 ± 5.6 nm (Figure 1h, 1i and Table S1).

Figure 1. (g) DLS analysis of BPNSs and BPNS-corona complexes. (h) DLS analysis of BPQDs and BPQD-corona complexes.

Table S1 DLS analysis of BP nanomaterials and BP-corona complexes

Nanomaterial	PBS		Serum-free Media	
	Particle size (nm)	PDI	Particle size (nm)	PDI
BPNSs	338.4 ± 2.3	0.28	341.2 ± 3.3	0.24
BPNS-corona complex	365.3 ± 5.9	0.19	359 ± 2.8	0.28
BPQDs	5.6 ± 1.4	0.22	5.8 ± 1.1	0.25
BPQDs-corona complex	362.5 ± 5.6	0.20	371.2 ± 4.7	0.31

Query 7

(7) Authors state that “BP nanosheets (BPNSs) were able to bind 25.0 percent of immune proteins from plasma, while BP quantum dots (BPQDs) bound 42.4 percent of immune proteins” since the absolute number of immune proteins in blood plasma is unknown, it would be more appropriate to state the total number of immune proteins identified, not the fraction.

Response

We thank the reviewer for this kind advice. We have revised manuscript to make it more clearly to state the total number of identified immune proteins in corona (Figure 2f). According to gene ontology analysis, 231 immune proteins in blood proteins, 97 immune proteins in BPQDs-cornua complexes and 54 immune proteins in BPNSs-cornua complexes were detected. The detail data about immune protein was shown in Table S2, S3 and S4. These results indicated that 41.8 percent of immune proteins from plasma were attracted to BPQDs and 23.3 percent of immune proteins were attracted to BPNSs (Figure 2g).

Revisions Made

(Please refer to page 7, line 120-123).

As shown in Figure 2f, 231 immune proteins in blood, 97 immune proteins in BPQD-cornua complexes and 54 immune proteins in BPNS-cornua complexes were detected. These results indicated that 41.8 percent of immune proteins from plasma were attracted to BPQDs, and 23.3 percent of immune proteins were attracted to BPNSs (Figure 2g).

Figure 2. Statistical analysis of the number of proteins on BPQDs and BPNSs involved in immune system processes GO analysis. Values are expressed as the means \pm SDs of triplicates. (g) Analysis of the immune protein fraction.

Query 8

(8) For zeta potential measurements, were the particles first separated from unbound protein? The unbound plasma proteins may interfere with light scattering measurements, particularly for small nanoparticles in the <20nm range, as is the case for BPQDs here. This should be made clear in the materials & methods section.

Response

We thank the reviewer for this kind advice. We have revised manuscript to make it more clearly about zeta potential measurements.

Revisions Made

(Please refer to page 17-18, line 309-315).

Zeta Potential and DLS Measurement. BP nanomaterials (BPQDs and BPNSs) were incubated with plasma at 4 °C for 4 h. TH PBS to thoroughly remove unbound plasma protein and centrifuged again for 20 min at

12000 rpm at 4 °C to form BP-complexes. After resuspension in PBS, the zeta potential and size of BP-corona complexes were measured on a Nano-ZS instrument (Malvern Instruments Limited) and dynamic light scattering (Brookhaven BI-200SM). In addition, the zeta potential and size of BPQDs and BPNSs were also measured in PBS. All experiments were conducted at least twice to ensure the reproducibility of the results.

Query 9

(9) In the methods section, plasma was isolated from different donors. Was this plasma then pooled? Since the donor is known to influence the composition of plasma, comparing samples incubated in plasma from different donors could lead to donor-dependent effects on the protein corona. This should be indicated explicitly in the manuscript.

Response

We thank the reviewer for the good question.

According to previous reports (Stauber *et al. ACS nano*, 5, 7155-7167 (2011); Dawson *et al. ACS nano*, 5, 7503-7509 (2011)), we collected blood from 10 different donors and then plasma was isolated by centrifuged. After that, plasma supernatant was collected and pooled together.

Please refer to page 16-17, line 284-288.

Blood was taken from 10 different apparently healthy donors (according to the Declaration of Helsinki) and stored in tubes containing EDTA to prevent blood clotting. The blood samples were labelled anonymously and could not be traced back to a specific donor. The samples were centrifuged for 5 min at 4000 rpm to pellet red and white blood cells. The plasma supernatant was pooled and stored at -80 °C. This plasma was used exclusively throughout the study.

Query 10

(10) Authors suggest that differences in the composition of the protein corona between BPNSs and BPQDs is due to nanoparticle size. However, such differences may also result from differences in the surface chemistry introduced during the manufacturing process. Would the authors make a reasoned argument as to why the observed effects are due to size and not surface chemistry?

Response

We thank the reviewer for the kind advice. To further study the surface chemistry, BPQDs and BPNSs were characterized by Raman spectroscopy and X-ray photoelectron spectroscopy (XPS) (Figure 1c and 1d). The Raman spectra results showed that, the three characteristic peaks can be due to one out-of plane phonon mode (A_g^1) at 356.8 cm^{-1} as well as two in-plane modes, B_{2g} and A_g^2 , at 433.2 cm^{-1} and 460.4 cm^{-1} , respectively. No prominent raman-shift was detected between BPNSs and BPQDs, indicating the oscillation of P atoms were similar on these two BP nanomaterials. Besides, the similar survey XPS spectra of BPQDs and BPNSs documented the similar surface chemistry of these two BP nanomaterials. Taken together, size is the key difference between BPQDs and BPNSs.

Revisions Made

(Please refer to page 3, line 54-59).

The BPQDs and BPNSs were also characterized by X-ray photoelectron spectroscopy (XPS) and Raman spectroscopy. As shown in Figure 1c, the similar prominent peaks from BPQDs and BPNSs can be attributed to one out-of-plane phonon mode (A_g^1) at 356.8 cm^{-1} as well as two in-plane modes, B_{2g} and A_g^2 , at 432.3 cm^{-1} and 459.6 cm^{-1} , respectively. Furthermore, similar survey XPS spectra for BPQDs and BPNSs document the similar surface chemistry of these two BP nanomaterials (Figure 1d). Taken together, the data indicate that size is the key difference between BPQDs and BPNSs.

Figure 1. (c) Raman spectra of BPQDs and BPNSs. (d) XPS spectra of BPQDs and BPNSs.

Query 11

(11) *Authors track intracellular localization of nanoparticle-protein complexes using a FITC-labeled BSA adsorbed to the surface. However, upon internalization (or even in the surrounding cell culture media), it is possible that the FITC-labeled BSA might desorb or be competitively displaced by protein in the surrounding media. In which case, in the fluorescence microscopy studies, the authors are simply observing free BSA and not BSA bound to nanoparticles. Additional studies should be undertaken to assess the stability of BSA-nanoparticle complexes in both plasma and artificial lysosomal fluid to verify that BSA remains attached to the particles under realistic physiological conditions.*

Response

We thank the reviewer for the good question. To analyse the stability of BP-corona-HSA-FITC complexes, we further tested the dissociated HSA-FITC in dTHP-1 cells lysate, which simulated the environment of macrophage cells. As shown in Figure S8, only 4.3% of HSA-FITC was dissociated from BPNS-corona complexes and 6.3% of HSA-FITC was dissociated from BPQD-corona complexes after 10 h, indicating the good stability of fluorescent corona complexes in intracellular environment.

Revisions Made

(Please refer to page 10, line 175-180).

First, we studied the stability of fluorescent corona complexes in macrophage cell lysates to simulate the macrophage cell environment. As shown in Figure S10, only 4.3% of HSA-FITC was dissociated from BPNS-corona complexes and 6.3% of HSA-FITC was dissociated from BPQD-corona complexes after 10 h, indicating the stability of fluorescent corona complexes in intracellular environments. In this way, we were able to label BP-corona complexes with fluorescence tags and determine their cellular location in dTHP-1 cells.

Figure S12. Stability of fluorescent BP-corona complexes. After remove the BP-corona-HSA-FITC complexes, the fluorescence of supernatant were detected at different time. The fluorescence of BP-corona-HAS-FITC suspensions was also measured as control. Values expressed were means \pm SDs of triplicates.

Query 12

(12) Authors carried out cell assays under serum-free conditions – which is not standard culture practice. Additional cell studies should be performed to study the effect of serum on cell uptake and serum-free conditions. In this case, nanoparticles could be pre-incubated with plasma and resuspended in either DMEM + 0% FBS or DMEM + 10% FBS and the effects compared.

Response

We thank the reviewer for the kind reminder. As previous lectures showed, the formation of corona on nanomaterials is a dynamic process, and the component of a human plasma corona are affected by fetal bovine serum (FBS) (Caruso *et al.* *ACS nano.* 7, 10960-10970 (2013); Dawson *et al.* *J. Am. Chem. Soc.* 132, 5761-5768 (2010)). Thus, it would be more suitable to carry out cell assays in serum-free media to eliminate the interference of FBS. To verify the effect of serum concentration on cell, we first performed an MTT assay to analyse the cell viability in different culture media (0% FBS and 10% FBS) after a short treatment. As Figure S8a show, after 6 h, the cell viabilities (without any treatment) are 98 % in serum-free media and 100% in serum media. Even treated with nanomaterials (100 µg/ml) in serum-free media, the cell viabilities are all over 90 %. We next compared the cellular uptake of BP-corona complexes in different culture medium (0% FBS and 10% FBS). The results of cellular uptake experiment showed that, FBS had no significant effect on cellular uptake efficiency during a short incubation period (Figure S8b). Besides, cell assays carried out in serum-free media could be observed in other reported literatures (Dawson and Monopoli *et al.* *ACS Nano.* 9, 2157-2166 (2015); Caruso *et al.* *ACS Nano.* 7, 10960-10970 (2013)). Based on the above results, we carried out further measurements using macrophages in serum-free media to eliminate the interference of FBS.

Revisions Made

(Please refer to page 8-10, line 147-165).

As previous literature has shown, the formation of a corona on nanomaterials is a dynamic process, and the components of a human plasma corona are affected by fetal bovine serum (FBS). Thus, it would be more suitable to carry out cell assays in serum-free media to eliminate the interference of FBS. To verify the effect of serum concentration on cells, we first performed an MTT assay to analyse the cell viability in different culture media (0% FBS and 10% FBS) after a short treatment. As Figure S8a shows, after 6 h, the cell viabilities (without any treatment) are 98% in serum-free media and 100% in serum media. Even treated with nanomaterials (100 µg/ml) in serum-free media, the cell viabilities are all over 90%. We next compared the cellular uptake of BP-corona complexes in different culture medium (0% FBS and 10% FBS). The results of the cellular uptake experiment showed that FBS had no significant effect on cellular

uptake efficiency during a short incubation period (Figure S8b). Based on the above results, we carried out further measurements using macrophages in serum-free media to eliminate the interference of FBS.

Figure S8. Effect of serum-free medium on macrophage cells. (a) Cytotoxicity of BP nanomaterials and corona complexes. Cells were treated with 100 µg/ml nanomaterials for 6 h in serum-free media. Medium with 0% FBS and 10% FBS were set as control. Values expressed were means ± SDs of triplicates. (b) Comparison of the cellular uptake of BP-corona complexes in different medium. Macrophage-like dTHP-1 cells were treated with 150 µg/ml corona complexes for 1, 3 and 6 h. Values expressed were means ± SDs of triplicates.

Query 13

(13) *The relevance of findings to other nanomaterial systems should be discussed.*

Response

We thank the reviewer for the good advice. Following this suggestion, we have searched literatures more carefully and revised our manuscript.

According to previous reports, the pro-inflammatory properties of different nanomaterials were linked with the ability to induce oxidative stress (Nel *et al. Nano Lett.* 6, 1794-1807 (2006); Rao *et al. Nat. Biotechnol.* 32, 373-380 (2014)). However, recent studies showed that the formation of protein corona was an alternative pathway for

nanomaterials to stimulate inflammation response (Wooley *et al. Chem. Soc. Rev.* 42, 5552-5576 (2013)). Caruso *et al.* have reported that protein corona formed by exposing nanoporous polymer particles (NPPs) to cell culture media, and found that the low-level of proinflammatory cytokine secretion and low-level early apoptosis were induced by NPPs-corona complexes (Caruso *et al. ACS nano.* 7, 10960-10970 (2013)). Moreover, as Minchin's report has shown, Poly(acrylic acid)-coated gold nanoparticles could unfold fibrinogen and provoke activation of Mac-1 receptor pathway in macrophage-like cells, indicating the potentially pro-inflammatory effect of fibrinogen-bound nanoparticles (Minchin *et al. Nat. Nanotechnol.* 6, 39-44 (2011)). Other nanomaterials system (i.e. iron oxide nanoparticles and SiO₂ nanoparticles) have also been reported (Dawson and Monopoli *et al. ACS Nano.* 9, 2157-2166 (2015); Simberg *et al. Nat. Nanotechnol.* 12, 387-393 (2017)).

Revisions Made

(Please refer to page 11, line 188-191).

Presently, increasing studies have shown that protein corona formation was an important pathway for nanomaterials to stimulate the inflammatory response. Other nanomaterial systems, including polymer nanoparticles (NPs), gold NPs, superparamagnetic iron oxide NPs, etc., have been reported to induce proinflammatory cytokines in immune cells.

Query 14

(14) Some typographical errors. Thorough proof-reading and copy-editing needed.

Response

We thank the reviewer for the kind reminder. Following this suggestion, we have carefully revised manuscript after proofread by two native speakers.

Reviewer: 2

In their manuscript entitled “Revealing the Immune Response of Black Phosphorus Nanomaterials by Understanding the Protein Corona” Mo et al. investigate the role of the protein corona on the physicochemical properties and biocompatibility of black phosphorous nanoparticles. (BPN)

While the protein corona of BPNs has not yet been investigated, there are currently several methodological, experimental and conceptual limitations that make it impossible to judge the validity of the presented data and preclude publication at this point. Data seem to be preliminary and insight into the current state of the art in nanoparticle proteomics and characterization of nanoparticle biology seems to be lacking.

1. What do the authors refer to by “the immune response of nanoparticles” - Many statements do not make any sense from an immunological perspective, e.g. Line 123 “Immune Response of BP Nanomaterials-corona Complexes” – do the authors refer to an immune response of the host against the nanomaterials? Investigating uptake of BPN and the cytokine secretion in serum free media of a single macrophage-like cell line is certainly not sufficient to characterize the host immune response. Any in vivo data are lacking.

Response

We thank the reviewer for the kind comments. The formation of corona on nanomaterials is quite complicated, and there is significant difference between human plasma and murine plasma. The human plasma corona formed on nanomaterials would be affected by murine plasma protein *in vivo*, and the component of human plasma corona would change. Basing on species-specific effects, it would be inappropriate to evaluate the immunological property of BP-human plasma corona complexes in murine cells or mouse model. Therefore, we have carried out all cellular experiments in human macrophages instead of mouse macrophages.

This study focused on the immunoregulation and immune perturbation effect of BP nanomaterials to macrophages via plasma corona. The main point of present study is to form BP-human plasma corona complexes *in vitro*, and analyses its impact on human macrophages (dTHP-1 and SC cells). Thus, we have carried out more assays to investigate the immunotoxicity and immune perturbation effect of BP corona complexes in human macrophages (Figure 4, 5 and S11). BP corona complexes could activate NF-

κ B pathway and increase cytokine generation. NO, TNF- α and ROS expression elicited by BP-corona complexes indicated the potential immunotoxicity of corona complexes. Furthermore, BP-corona complexes could affect the response of macrophages to foreign materials. It is our ongoing work to study the immunological properties of BP-murine plasma corona complexes *in vivo*.

Taken together, we have carefully demonstrated the immunoregulation effect of BP corona complexes on human macrophages and revised manuscript to make it more accurate in expression.

Revisions Made

(Please refer to page 11-15, line 188-251).

Proinflammatory Effect of BP-corona Complexes in Macrophages. Presently, increasing studies have shown that protein corona formation was an important pathway for nanomaterials to stimulate the inflammatory response. Other nanomaterial systems, including polymer nanoparticles (NPs), gold NPs, superparamagnetic iron oxide NPs, etc., have been reported to induce proinflammatory cytokines in immune cells. To test whether the protein corona bound to BP nanomaterials could increase the secretion of cytokines, macrophage cells were treated with either BP nanomaterials or BP-corona complexes. Subsequently, the culture supernatants were collected, and the amount of inflammatory cytokines was detected by enzyme-linked immunosorbent assay (ELISA). Significant changes in cytokine secretion were observed in macrophage cells (dTHP-1 cells and SC cells) treated with corona complexes. As shown in Figure 4a, BP-corona complexes induced an obvious increase in 4 inflammatory cytokines: interleukin (IL)-1 β , IL-6, IL-8, and interferon (IFN)- γ . In particular, these cytokine secretions in dTHP-1 cells were increased 2-4-fold above background level. However, compared with BP-corona complexes, BP nanomaterials had no significant effect on the secretion of cytokines. IL-9 and IL-10 expression was also not obviously changed upon exposure to BP nanomaterials or BP-corona complexes. Moreover, a similar trend in cytokine expression was observed in macrophages from peripheral blood (SC cells). The secretion of IL-1 β , IL-6, IL-8, IL-9 and IL-10 in SC cells was also enhanced by BP-corona complexes (Figure 4b). However, IFN- γ expression was not significantly changed upon exposure to BP nanomaterials or corona complexes. Taken together, the higher secretion of inflammatory cytokines observed in different human macrophage

cells are in support of the proinflammatory effect of BP nanomaterials in the presence of a plasma corona.

Generally, the NF- κ B pathway has been implicated as a key factor in inflammatory effects induced by foreign materials. The activation of NF- κ B results in I κ B degradation and nuclear translocation of the p65 dimer. After treatment with BP nanomaterials or corona complexes, the cytoplasmic and nuclear extracts of dTHP-1 cells were analysed by Western blot. The results in Figure S11 show I κ B degradation and nuclear translocation of p65. These changes elicited by BP nanomaterials were much lower than those observed with corona complexes. To further understand the relationship between immune proteins on BP nanomaterials and immune effects, we carefully searched the reported literatures for the effect of every immune protein identified by GO analysis and speculated that BP-corona complexes have more potential immune responses (Table S5).

Immune Perturbation of BP-corona Complexes in Macrophages. Previous reports have shown that the induction of nitric oxide (NO) and tumour necrosis factor (TNF)- α production is a molecular marker for the immunotoxicity of nanomaterials. To elucidate the immune perturbation in macrophages, we first studied the induction effect of BP-corona complexes on NO and TNF- α . As shown in Figure 5a, TNF- α and NO production were provoked by BP nanomaterials and corona complexes. The increase in TNF- α and NO expression induced by corona complexes was 2 times greater than that of BP nanomaterials. The bacterial endotoxin LPS is known to induce the release of inflammatory factors in macrophages. To analyse the effect of corona complexes on macrophages function, we further evaluated stress in macrophages caused by foreign materials. As the results of Figure 5b show, after pretreatment with BP-corona complexes, the release of TNF- α in dTHP-1 cells treated with LPS was significantly inhibited. For example, after dTHP-1 cells were pretreated with 50 μ g/ml BPQD-corona complexes and then incubated with 20 ng/ml LPS, the secretion of TNF- α was decreased to 49.83 pg/ml from 95.16 pg/ml (without pretreatment with BPQD-corona complexes). These results supported the immunotoxicity of BP-corona complexes to macrophages.

Reactive oxygen species (ROS) are an important part of the primary immune defence in macrophages against foreign materials. In addition, the expression of ROS has also been recognized as a major toxicological paradigm of nanomaterials. The ROS level induced by BP-corona complexes was tested by dichlorofluorescein (DCF) fluorescence

assay (Figure 5c). ROS production was significantly increased in a time-dependent manner in macrophage cells treated with BP nanomaterials or BP-corona complexes. Compared with BP nanomaterials, corona complexes induced much higher ROS production. For example, BPQD- and BPNS-corona complexes triggered ROS overproduction in dTHP-1 cells at approximately 185.8% and 215.2% at 150 min, which was much higher than that elicited by BPQDs (143.7%) and BPQNs (159.9%). These findings demonstrated that BP nanomaterials exhibited enhanced immune perturbations in macrophages because of the formation of plasma coronas.

a)

b)

Figure 4. Cytokine secretion of macrophage-like (a) dTHP-1 cells and (b) SC cells. Cells were treated with 50 $\mu\text{g/ml}$ BP nanomaterials and an increasing concentration of corona complexes (12.5, 25 and 50 $\mu\text{g/ml}$) for 6 h. Values are expressed as the means \pm SDs of triplicates. * $p < 0.05$, ** $p < 0.01$

Figure 5. Immune perturbations of BP-corona complexes in macrophages. (a) NO and TNF- α generation in dTHP-1 cells. Cells were treated with 50 $\mu\text{g/ml}$ BP nanomaterials and an increasing concentration of corona complexes (12.5, 25 and 50 $\mu\text{g/ml}$) for 6 h. Values are expressed as the means \pm SDs of triplicates. * $p < 0.05$, ** $p < 0.01$. (b) TNF- α secretion in dTHP-1 cells. LPS (20 ng/ml) was added to the macrophages, which had been pretreated with BP nanomaterials or BP-corona complexes for 6 h. Values are expressed as the means \pm SDs of triplicates. (c) ROS overproduction in macrophages exposed to 50 $\mu\text{g/ml}$ BP nanomaterials or BP-corona complexes for 6 h. LPS (20 ng/ml) were set as the positive control. Values are expressed as the means \pm SDs of triplicates.

Figure S11. Activation of NF-κB pathways by BP nanomaterials or corona complexes. Macrophage-like dTHP-1 cells were treated with nanomaterials at 100 µg/ml for 6h.

2. *The proteomics data are insufficiently described. No data are provided except on a highly aggregated level. The entire dataset is not presented (neither in the manuscript nor supplementary information), making it impossible to judge the validity of results. No information is provided regarding number of biological or technical replicates for the proteomic analyses. The authors must provide the entire datasets in the supplementary information, data should include (at least): Protein Identifier, Scores, FDR, number of peptides per proteins, quantification values from all biological and technical replicates. Methods for proteomic analysis are insufficiently described. No details regarding lysis buffers or LC conditions are provided. No details for search parameters are provided. Please refer to MCP reporting guidelines for proper reporting of proteomic datasets. Additionally, I highly recommend submitting the datasets (rawdata and search results) to a public repository (i.e. ProteomeXchange) to allow other researchers to access the data.*

Response

We thank the reviewer for the kind comments. Following this suggestion, we have carefully checked the “Identification of Protein Corona” section. Protein identification experiment carried out in this study was qualitative analysis rather than quantitative analysis. Besides, we have provided the qualitative proteomics data in supplementary information (Table S2, S3 and S4). The data include: Uniprot accession number, Protein

ID, PepCount, UniquePepCount, CoverPercent and Score. CoverPercent could be used to indirectly evaluate the relative abundance of identified protein in corona. The proteomic data obtained from technical replicates was also provided in supplementary information. Finally, we have carefully revised manuscript to make it more clearly about the methods for proteomic analysis. The datasets have been submitted to the *iProX* (accession number: IPX0001162000) and *ProteomeXchange* (accession number: PXD009126) database.

Revisions Made

(Please refer to page 7, line 113-114).

Next, we used liquid chromatography tandem mass spectrometry (LC-MS/MS) to qualitatively analyse components of the BP-protein corona (Table S2, S3 and S4).

(Please refer to page 18-19, line 321-343).

Protein Identification and Classification. The nanoparticle-protein complexes were collected by the above methods. Corona complexes were depleted of the most abundant proteins using an Agilent Human 14 Multiple Affinity Removal System Column following the manufacturer's protocol. Then, one volume of SDT buffer (4% SDS, 100 mM DTT, 150 mM Tris-HCl pH 8.0) was added, and the solution was boiled for 15 min and centrifuged at 14000 g for 20 min. After that, 200 µg of proteins from each sample was incorporated into 30 µl SDT buffer. The detergent, DTT and other low-molecular-weight components were removed using UA buffer (8 M urea, 150 mM Tris-HCl pH 8.0) by repeated ultrafiltration (Microcon units, 10 kDa). Then, 100 µl iodoacetamide (100 mM IAA in UA buffer) was added to block reduced cysteine residues, and the samples were incubated for 30 min in the dark. The filters were washed with 100 µl UA buffer three times and then 100 µl 25 mM NH₄HCO₃ buffer twice. Finally, the protein suspensions were digested with 4 µg trypsin (Promega) in 40 µl 25 mM NH₄HCO₃ buffer overnight at 37 °C, and the resulting peptides were collected as a filtrate. The peptides from each sample were desalted on C18 Cartridges (Empore™ SPE Cartridges C18, bed I.D. 7 mm, volume 3 ml, Sigma), concentrated by vacuum centrifugation and reconstituted in 40 µl of 0.1% (v/v) formic acid (Fluka). The peptide content was estimated by UV light spectral density at 280 nm using an extinction

coefficient of 1.1 of 0.1% (g/l) solution that was calculated on the basis of the frequency of tryptophan and tyrosine in vertebrate proteins. Then, each fraction was injected into a Q Exactive mass spectrometer (Thermo Scientific) for nanoLC-MS/MS analysis. MS/MS spectra were searched using the MASCOT engine (Matrix Science, London, UK; version 2.4). For protein identification, the following options were used: peptide mass tolerance=20 ppm, MS/MS tolerance=0.1 Da, enzyme=Trypsin, missed cleavage=2, fixed modification: carbamidomethyl (C), and variable modification: oxidation (M). The identified proteins were retrieved from the UniProtKB human database (Release 2017_02) in FASTA format. In this study, the top 10 blast hits with E-values less than 1e-3 for each query protein were retrieved and loaded into Blast2GO (Version 3.3.5) for GO mapping and annotation. The GO analysis was carried out according to previous literature. All experiments were conducted at least twice to ensure the reproducibility of the results.

3. The authors' interpretations of the proteomics data are strange. The authors only look at numbers of proteins and actual quantification results are hidden from the reader. What is actually meant by "discovered that 42.4 percent of immune proteins from plasma were attracted to BPQDs and 25 percent of immune proteins were attracted to BPNSs"?

Response

We thank the reviewer for the good question. According to the reviewer's comments, we have revised the manuscript to make it more clearly to state the total number of identified immune proteins in corona (Figure 2f). According to gene ontology analysis, 231 immune proteins in plasma proteins, 97 immune proteins in BPQD-corona complexes and 54 immune proteins in BPNS-corona complexes were detected. These results indicated that 41.8 percent of immune proteins from plasma were attracted to BPQDs and 23.3 percent of immune proteins were attracted to BPNSs (Figure 2g).

Revisions Made

(Please refer to page 7, line 120-123).

As shown in Figure 2f, 231 immune proteins in blood, 97 immune proteins in BPQD-corona complexes and 54 immune proteins in BPNS-corona complexes were detected. These results indicated that 41.8 percent of immune proteins from plasma were attracted to BPQDs, and 23.3 percent of immune proteins were attracted to BPNSs (Figure 2g).

Figure 2. Statistical analysis of the number of proteins about immune system process on BPQDs and BPNSs according to GO analysis. Values are expressed as the means \pm SD of triplicate. (g) Analysis of the fraction of immune protein.

4. Most recent contributions by other groups concerning the role of the protein corona on nanoparticle biology have been ignored by the authors.

Response

We thank the reviewer for the good question. Following this suggestion, we have searched literatures more carefully and revised our manuscript accordingly.

In addition to the low-level damage in cell membranes and loss of targeting capabilities, protein corona can modulate other nanoparticle-induced biological processes. Stauber *et al.* have revealed that rapid corona formation could affect haemolysis, thrombocyte activation and endothelial cell death (Stauber *et al. Nat. Nanotechnol.* 8, 772-781 (2013)). Lin *et al.* have also reported that coating of polymeric nanoparticles with human serum albumin could be used to increase their blood circulation time (Lin *et al. Biomaterials.* 34, 8521-8530 (2013)). Besides, other roles of protein corona on nanomaterials have also been reported, including alteration in organ distribution,

immunostimulatory and immunosuppressive effect (Muller *et al.* *J. Drug Target.* 13, 179-187 (2005); McNeil *et al.* *Nat. Nanotechnol.* 2, 469-478 (2007)).

Revisions Made

(Please refer to page 15, line 254-256).

So far, different roles of nanomaterial-protein corona have been defined, including reduction of toxicity to cells, loss of targeting capabilities, extension of blood circulation time, alterations in organ distribution, and immunostimulatory and immunosuppressive effects.

5. Incubation of nanoparticles with serum/plasma at 4°C is a highly artificial setting – why were the experiments not performed at 37°C?

Response

We thank the reviewer for the good question.

According to previous literature, different incubation temperatures were adopted to obtain nanoparticle-corona complexes (4 °C Dawson *et al.* *J. Am. Chem. Soc.* 133, 2525-2534 (2011); Suslick *et al.* *Nano Lett.* 14, 6-12 (2014); Puustinen *et al.* *ACS nano.* 5, 4300-4309 (2011). 37 °C. Huang *et al.* *ACS Nano.* 5, 3693-3700 (2011); Chan *et al.* *Angew. Chem.* 53, 5093-5096 (2014); Minchin *et al.* *Nat. Nanotechnol.* 6, 39-44 (2011)). For analyse the difference of corona obtained from different incubation temperature, we have incubated the BP nanomaterials with plasma at 37 °C and identified the component in corona by LC-MS/MS. Subsequently, the identified protein was further classified by gene ontology analysis according to their biological process. The results in the following Figure (data no showed in manuscript) showed a similar trend in component of corona between different incubation temperatures. These results documented that incubation temperature is an inessential factor in the formation of BP-corona.

Figure. Effect of incubation temperature on the immune protein component of BP-corona complexes.

6. The cytokine secretion assay was performed in serum-free media, which typically leads to starvation and subsequent extensive cell death.

Response

We thank the reviewer for the good question. As previous reports suggested, the formation of corona on nanomaterials is a dynamic interaction, and the component of human plasma-corona would be affected by fetal bovine serum (FBS) (Caruso *et al. ACS nano.* 7, 10960-10970 (2013); Dawson *et al. J. Am. Chem. Soc.* 132, 5761-5768 (2010)). Thus, it would be more suitable to carry out cell assays in serum-free media to eliminate the interference of FBS. Besides, cytokine level was measured after macrophages treated with corona complexes for 6 h. To verify the effect of serum-free condition on cell viability, we performed an MTT assay to analyse the cell viability in different culture media (0% FBS and 10% FBS) after a short treatment. As Figure S8a shows, after 6 h, the cell viabilities (without any treatment) are 98% in serum-free media and 100% in serum media. Even treated with corona complexes (100 µg/ml) in serum-free media, the cell viabilities are all over 90%. Besides, cell assays carried out in serum-free media could be found in other reported literatures (Dawson and Monopoli *et al. ACS Nano.* 9, 2157-2166 (2015); Caruso *et al. ACS Nano.* 7, 10960-10970 (2013)).

Based on the above results, we carried out further measurements using macrophages in serum-free media to eliminate the interference of FBS.

Revisions Made

(Please refer to page 8-10, line 147-165).

As previous literature has shown, the formation of a corona on nanomaterials is a dynamic process, and the components of a human plasma corona are affected by fetal bovine serum (FBS). Thus, it would be more suitable to carry out cell assays in serum-free media to eliminate the interference of FBS. To verify the effect of serum concentration on cells, we first performed an MTT assay to analyse the cell viability in different culture media (0% FBS and 10% FBS) after a short treatment. As Figure S8a shows, after 6 h, the cell viabilities (without any treatment) are 98% in serum-free media and 100% in serum media. Even treated with nanomaterials (100 $\mu\text{g}/\text{ml}$) in serum-free media, the cell viabilities are all over 90%. We next compared the cellular uptake of BP-corona complexes in different culture medium (0% FBS and 10% FBS). The results of the cellular uptake experiment showed that FBS had no significant effect on cellular uptake efficiency during a short incubation period (Figure S8b). Based on the above results, we carried out further measurements using macrophages in serum-free media to eliminate the interference of FBS.

Figure S8. Effect of serum-free medium on macrophage cells. (a) Cytotoxicity of BP nanomaterials and corona complexes. Cells were treated with 100 $\mu\text{g}/\text{ml}$ nanomaterials for 6 h in serum-free media. Medium with 0% FBS and 10% FBS were set as control. Values expressed were means \pm SDs of triplicates. (b) Comparison of the cellular

uptake of BP-corona complexes in different medium. Macrophage-like dTHP-1 cells were treated with 150 $\mu\text{g/ml}$ corona complexes for 1, 3 and 6 h. Values expressed were means \pm SDs of triplicates.

Reviewers' Comments:

Reviewer #1:

Remarks to the Author:

The authors have adequately addressed each of the concerns raised in my original review. The overall quality of the manuscript has been significantly improved and the data and analyses now support the central claims. This manuscript is suitable for publication in Nature Communications. In addition, I believe it will make a valuable addition to the literature focused on understanding the interaction of inorganic nanomaterials with biological systems in general and the biological application of black phosphorus nanoparticles in particular.

Reviewer #2:

Remarks to the Author:

In their manuscript entitled „Revealing the immune perturbation of black phosphorus nanomaterials to macrophages by understanding the protein corona“ Mo et al investigate the role of the protein corona on black phosphorous nanosheets.

While the cell culture and biochemical part of the manuscript has significantly improved, the mass spectrometric data are still highly questionable, which in my view precludes publication at this point. There are some inconsistencies in the materials and methods and data should be critically re-evaluated to eliminate potential false positive identifications ("single peptide hits")

There are a number of issues:

1)

Page 1 "we discovered that BP nanosheets (BPNSs) were able to bind 23.3 percent of immune proteins from plasma, while BP quantum dots (BPQDs) bound 41.8 percent of immune proteins"

What are "percent of immune proteins"? This is not a commonly used term. Please explain? I would suggest e.g. the wording " XX% of the proteins bound to BPNS were annotated as immune relevant proteins" (if this is what the authors wanted to express)

2) How do the authors define "immunotoxicity"? The particles kill macrophage-like cell lines in an in vitro system. This is not an indication of relevant in vivo toxicity.

3) The mass spectrometric methods are questionable. The number of reported proteins includes a significant number of single peptide hits, most of which were detectable only in a single analysis, which is highly indication of false positive identifications.

All proteins must be filtered to report only proteins with at least two identified peptides, ideally in two replicate analyses.

According to the materials and methods, no FDR filtering was performed. Which kind of decoy strategy did the authors use to control false-positive IDs?

An MS/MS tolerance of 0.1Da is excessively large for Q-Exactive data, which might lead to false positive identifications. Please explain.

4) In the materials and methods, the authors wrote "Corona complexes were depleted of the most abundant proteins using an Agilent Human 14 Multiple Affinity Removal System Column following the manufacturer's protocol. "

To my knowledge, MARS columns require significant amount of input material.

How did the authors apply corona complexes to the column?

This strategy is highly questionable, as the MARS columns remove e.g. albumin almost quantitatively – so any particle with bound albumin (or any other of the 14 target proteins) would

be retained on the column.

Notably, the top hits in Table S2 (the proteins supposedly identified after depletion) are target proteins of the MARS column, further questioning the validity of the data or method descriptions.

Minor points:

The authors must always differentiate between macrophages (e.g ex-vivo generated) and macrophage-like cell lines.

Responses to Referee's Comments

Reviewer 1

The authors have adequately addressed each of the concerns raised in my original review. The overall quality of the manuscript has been significantly improved and the data and analyses now support the central claims. This manuscript is suitable for publication in Nature Communications. In addition, I believe it will make a valuable addition to the literature focused on understanding the interaction of inorganic nanomaterials with biological systems in general and the biological application of black phosphorus nanoparticles in particular.

Response

We thank the reviewer for this very positive comment.

Reviewer: 2

In their manuscript entitled "Revealing the immune perturbation of black phosphorus nanomaterials to macrophages by understanding the protein corona" Mo et al investigate the role of the protein corona on black phosphorus nanosheets.

While the cell culture and biochemical part of the manuscript has significantly improved, the mass spectrometric data are still highly questionable, which in my view precludes publication at this point. There are some inconsistencies in the materials and methods and data should be critically re-evaluated to eliminate potential false positive identifications ("single peptide hits")

1. Page 1 "we discovered that BP nanosheets (BPNSs) were able to bind 23.3 percent of immune proteins from plasma, while BP quantum dots (BPQDs) bound 41.8 percent of immune proteins"

What are "percent of immune proteins"? This is not a commonly used term. Please explain?

I would suggest e.g. the wording “XX% of the proteins bound to BPNS were annotated as immune relevant proteins” (if this is what the authors wanted to express)

Response

We thank the reviewer for this kind advice. Following this suggestion, we have carefully revised manuscript. As shown in Figure 2f, 186 proteins in plasma, 73 proteins in BPQD-corona complexes and 36 proteins in BPNS-corona complexes were annotated as immune relevant proteins. These results indicate that 75.8% of the proteins bound to BPQDs were immune relevant proteins, while the proportion of BPNS-corona complexes is 69.9% (Figure 2g).

Revisions Made

(Please refer to page 7, line 114-125).

Next, we used liquid chromatography tandem mass spectrometry (LC-MS/MS) to qualitatively analyse components of the BP-protein corona (Supplementary Table 2, 3 and 4). Further analysis with LC-MS/MS revealed that while BPNSs were able to adsorb 52 plasma proteins, BPQDs were capable of binding 96 proteins. Of these proteins, 16 proteins appeared on both BPNSs and BPQDs (Figure 2e). Subsequently, we analysed the difference in protein component among plasma proteins, BPQD-corona complex proteins and BPNS-corona complex proteins. First, this analysis showed that the proteins on BPQDs and BPNSs do not simply correspond to the relative protein concentrations in blood plasma, as reported by other studies that focused on other types of nanomaterials^{16, 18}. The bound proteins were further classified by GO analysis according to their biological process (Supplementary Figure 3). As shown in Figure 2f, of 186 proteins in blood, 73 proteins in BPQD-corona complexes and 36 proteins in BPNS-corona complexes were annotated as immune relevant proteins. These results indicated that 75.8% of the proteins bound to BPQDs (96 plasma proteins) were immune relevant proteins, while that percentage for BPNS-corona complexes (52 plasma proteins) is 69.9% (Figure 2g).

Figure 2. (f) the number of proteins on BPQDs and BPNSs involved in immune system processes according to gene ontology (GO) analysis. Values are expressed as the means \pm SDs of triplicates. (g) Analysis of the immune relevant protein fraction.

2. How do the authors define “immunotoxicity”? The particles kill macrophage-like cell lines in an *in vitro* system. This is not an indication of relevant *in vivo* toxicity.

Response

We thank the reviewer for this suggestion and accordingly, we have carefully revised the manuscript to make the description of immunotoxicity of corona complexes more precise. According to previous reports, immunotoxicity of nanomaterials refers to the effect of nanomaterials on immune cell functions, and includes secretion of inflammatory cytokines, ROS overexpression, phagocytic activity and proliferative activity (Dusinska et al. *Nanotoxicology*. 9, 33-43 (2013); Santos et al. *Biomaterials*. 34, 7776-7789 (2013); Cai et al. *J. Biomed. Mater. Res. Part A*. 102, 3781-3794 (2015)). Evaluation *in vitro* of the immunotoxicity of nanomaterials has been reported. According to our results, BP-corona complexes did not kill macrophage-like cells, but the cytokine secretion, ROS expression, nitric oxide production and stress in macrophages were all perturbed by BP-corona complexes. Based on published reports and our results, we indicated that BP-corona complexes showed immunotoxicity to macrophages *in vitro*.

Revisions Made

(Please refer to page 14, line 233-236).

According to previous reports, immunotoxicity of nanomaterials means the effect of nanomaterials on immune cell functions, including inflammatory cytokine secretions, ROS overexpression, phagocytic activity, proliferative activity, etc³¹⁻³³. Evaluation *in vitro* of the Immunotoxicity of nanomaterials has been reported^{34, 35}. In addition, the induction of nitric oxide (NO) and tumour necrosis factor (TNF)- α production are molecular markers of the immunotoxicity of nanomaterials^{36, 37}.

3. The mass spectrometric methods are questionable. The number of reported proteins includes a significant number of single peptide hits, most of which were detectable only in a single analysis, which is highly indication of false positive identifications.

All proteins must be filtered to report only proteins with at least two identified peptides, ideally in two replicate analyses.

According to the materials and methods, no FDR filtering was performed. Which kind of decoy strategy did the authors use to control false-positive IDs?

An MS/MS tolerance of 0.1Da is excessively large for Q-Exactive data, which might lead to false positive identifications. Please explain.

Response

We thank the reviewer for this good question. In the light of the reviewer's comments, we have carefully revised the analysis of the mass spectrometric data. We eliminated the proteins with single peptide hits, and retained the detected proteins with two or more identified peptides. As shown by the results in Figures 2e and 2f, the numbers of proteins with at least two identified peptides in plasma, BPQD- and BPNS-corona complexes are 260, 96 and 52, respectively. Furthermore, the numbers of immune relevant proteins with at least two identified peptides, in plasma, BPQD- and BPNS-corona complexes are 186, 73 and 36, respectively.

To control false-positive IDs, we set mascot score of >20 as a threshold. This decoy strategy has been reported in previous literatures (Heck *et al.* Nat. Protoc. 7, 2041-2055 (2012); Meisinger *et al.* Mol. Cell. Proteomics.11, 1840-1852 (2012); Deng *et al.* Autophagy. 13, 1318-1330 (2017); Podtelejnikov *et al.* Mol. Cell. Proteomics. 3, 1023-1038 (2004)).

Although an MS/MS tolerance of 0.1 Da was used for protein identification, the value of Diff (MH+) (the difference between the actual molecular weight and the theoretical molecular weight of the identified peptides) was less than ± 0.01 . Consequently, an MS/MS tolerance of 0.1 Da will not lead to false positive identifications. This method of protein identification has been reported (Acuto *et al. Mol. Cell. Proteomics.* 11, 1489-1499 (2012); Farese *et al. Mol. Cell. Proteomics.* 2018: mcp.RA117.000560; Pagliarini *et al. Mol. Cell. Proteomics.* 12, 3360-3369 (2013); Wu *et al. ACS nano.* 11, 3690-3704 (2017); Du *et al. Mol. Cell. Proteomics.* 15, 266-288 (2016)).

Revisions Made

(Please refer to page 7, line 114-117).

Next, we used liquid chromatography tandem mass spectrometry (LC-MS/MS) to qualitatively analyse components of the BP-protein corona (Supplementary Table 2, 3 and 4). Further analysis with LC-MS/MS revealed that while BPNSs were able to adsorb 52 plasma proteins, BPQDs were capable of binding 96 proteins. Of these proteins, 16 proteins appeared on both BPNSs and BPQDs (Figure 2e).

(Please refer to page 7, line 122-125).

The bound proteins were further classified by GO analysis according to their biological process (Supplementary Figure 3). As shown in Figure 2f, of 186 proteins in blood, 73 proteins in BPQD-corona complexes and 36 proteins in BPNS-corona complexes were annotated as immune relevant proteins. These results indicated that 75.8% of the proteins bound to BPQDs (96 plasma proteins) were immune relevant proteins, while that percentage for BPNS-corona complexes (52 plasma proteins) is 69.9% (Figure 2g).

Figure 2. (e) Statistical analysis of the total number of proteins identified by LC-MS/MS and (f) the number of proteins on BPQDs and BPNSs involved in immune system processes according to gene ontology (GO) analysis. Values are expressed as the means \pm SDs of triplicates. (g) Analysis of the immune relevant protein fraction.

4. In the materials and methods, the authors wrote “Corona complexes were depleted of the most abundant proteins using an Agilent Human 14 Multiple Affinity Removal System Column following the manufacturer’s protocol. “

To my knowledge, MARS columns require significant amount of input material.

How did the authors apply corona complexes to the column?

This strategy is highly questionable, as the MARS columns remove e.g. albumin almost quantitatively – so any particle with bound albumin (or any other of the 14 target proteins) would be retained on the column.

Notably, the top hits in Table S2 (the proteins supposedly identified after depletion) are target proteins of the MARS column, further questioning the validity of the data or method descriptions.

Response

We thank the reviewer for pointing out our error. We have carefully checked the “Protein Identification and Classification.” section. We realized that this was a reporting error because we did not actually use MARS columns in our study. Thus, we revised the description of the method for protein identification accordingly.

Revisions Made

(Please refer to page 18-19, line 330-341).

The BP-corona complexes were collected by the above methods. Then, one volume of SDT buffer (4% SDS, 100 mM DTT, 150 mM Tris-HCl pH 8.0) was added, and the solution was boiled for 15 min and centrifuged at 14000 g for 20 min. Digestion of protein (200 μ g for each sample) was performed according to the FASP procedure described by Mann *et al*⁴⁹. Briefly, the detergent, DTT and other low-molecular-weight components were removed using 200 μ l UA buffer (8 M Urea, 150 mM Tris-HCl pH 8.0) by repeated ultrafiltration (Microcon units, 30 kD) facilitated by centrifugation. Then 100 μ l

iodoacetamide (0.05 M in UA buffer) was added to block reduced cysteine residues and the samples were incubated for 20 min in the dark. The filter was washed with 100 μ l UA buffer three times and then 100 μ l 25 mM NH_4HCO_3 twice. Finally, the protein suspension was digested with 3 μ g trypsin (Promega) in 40 μ l 25 mM NH_4HCO_3 overnight at 37 °C, and the resulting peptides were collected as a filtrate. The peptide content was estimated by UV light spectral density at 280 nm using an extinction coefficient of 1.1 of 0.1% (g/l) solution, calculated on the basis of the frequency of tryptophan and tyrosine in vertebrate proteins.

5. The authors must always differentiate between macrophages (e.g ex-vivo generated) and macrophage-like cell lines.

Response

We thank the reviewer for this kind advice. Following the reviewer's suggestion, we have carefully revised the manuscript to differentiate between macrophages and macrophage-like cells.

Revisions Made

(Please refer to page 10, line 167; page 11, line 187; page 11, line 195; page 14, line 243; page 15, line 253).

Reviewers' Comments:

Reviewer #2:

Remarks to the Author:

The authors have sufficiently addressed the remaining points. The manuscript is now suitable for publication.